# A new initialisation to Control Gradients in Sinusoidal Neural network

**Andrea Combette, Antoine Venaille & Nelly Pustelnik**
CNRS, ENS de Lyon
Laboratoire de Physique UMR5672
Lyon, 69007, France
`{andrea.combette,antoine.venaille,nelly.pustelnik}@ens-lyon.fr`

## Abstract

Proper initialisation strategy is of primary importance to mitigate gradient explosion or vanishing when training neural networks. Yet, the impact of initialisation parameters still lacks a precise theoretical understanding for several well-established architectures. Here, we propose a new initialisation for networks with sinusoidal activation functions such as `SIREN`, focusing on gradients control, their scaling with network depth, their impact on training and on generalization. To achieve this, we identify a closed-form expression for the initialisation of the parameters, differing from the original `SIREN` scheme. This expression is derived from fixed points obtained through the convergence of pre-activation distribution and the variance of Jacobian sequences. Controlling both gradients and targeting vanishing pre-activation helps preventing the emergence of inappropriate frequencies during estimation, thereby improving generalization. We further show that this initialisation strongly influences training dynamics through the Neural Tangent Kernel framework (NTK). Finally, we benchmark `SIREN` with the proposed initialisation against the original scheme and other baselines on function fitting and image reconstruction. The new initialisation consistently outperforms state-of-the-art methods across a wide range of reconstruction tasks, including those involving physics-informed neural networks.

## 1 Introduction

### 1.1 Context and Motivation

Implicit neural representations (INRs) have become a prevalent tool for approximating functions in diverse applications, including signal encoding (Strümpler et al., 2022; Dupont et al., 2021), signal reconstruction (Park et al., 2019; Mildenhall et al., 2020), and solutions of partial differential equations (PDEs) (Raissi et al., 2019). A central challenge in these neural approximations is to recover the frequency spectrum of a target signal within reasonable training time and from limited data. In this context, standard multi-layer perceptrons (MLPs) used for INRs often suffer from *spectral bias*, whereby low-frequency components are preferentially learned compared to high-frequency details (Rahaman et al., 2019; Li et al., 2024). This bias can hinder performance, either by slowing training or by reducing precision, when the signal of interest contains significant high-frequency content (fine textures, details, . . . ). To mitigate this issue, several architectures have been proposed, such as positional encoding (Tancik et al., 2020) or networks with sinusoidal activation functions (`SIREN`, (Sitzmann et al., 2020)), which enable faster learning of high-frequency components. However, increasing network depth in these methods has been empirically observed to introduce in the reconstructed function spurious high-frequency components absent from the target one (see, e.g., (Ma et al., 2025)), leading to noisy representations and degraded generalization (i.e., the ability to interpolate the signal correctly).

In this work, we propose an initialisation strategy for `SIREN` that bypasses two opposing pitfalls: (i) slow training and poor recovery of high-frequency details due to spectral bias in standard MLPs, and (ii) rapid training in deeper `SIREN`, which comes at the cost of spurious high-frequency artifacts and degraded generalization. Finding the right balance between these two extremes corresponds

to locating the frontier between vanishing-gradient and exploding-gradient regimes. Operating in this regime, where gradients remain stable, is often referred to as computing at the edge of chaos (Yang & Schoenholz, 2017; Seleznova & Kutyniok, 2022), a concept from dynamical systems theory (Kelso et al., 1986; Langton, 1986). Building on this idea, we introduce an explicit initialisation scheme for SIREN. Our method ensures that inputs and parameters gradients neither vanish nor explode with depth enabling both stable and expressive learning dynamics. With appropriate tuning of the pre-activation statistics it allows to impose a finite range of frequency at initialisation, allowing the network to capture high-frequency contents without introducing spurious components.

To better understand the critical role of the initialisation in INRs, we complement our theoretical analysis with experiments based on the neural tangent kernel (NTK) framework (Jacot et al., 2018; Li et al., 2024). We find that controlling gradient propagation at initialisation strongly influences the NTK eigenvalues, which determine the training speed of the frequencies associated with the corresponding eigenvectors.

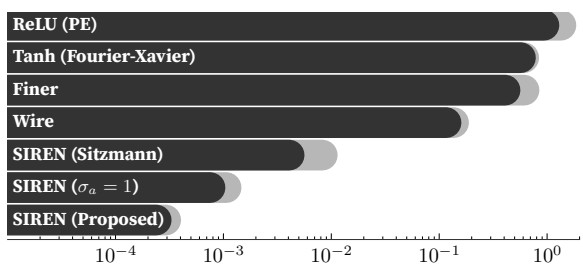

Figure 1: Generalization error over different problems averaged over different architecture depths for 1d, 2d and 3d multi-scale function approximation. The results are displayed for different state-of-the-art architectures including the one proposed in this work (**SIREN Proposed**). See Appendix B.6.3 for details. In standard deviation of the error is colored in light gray.

Beyond the NTK framework, our initialisation prevents the degradation of deep neural network performance with increasing depth. We illustrate this property across several function fitting problem in Figure 1 where a comparison of the performance of our initialisation against the original SIREN scheme and other baselines is presented.

## 1.2 RELATED WORK

**Frequency representation.** Our study will be based on the work of (Sitzmann et al., 2020), which introduced the SIREN architecture, a neural network with sinusoidal activation functions designed to effectively learn high-frequency functions by using a tunable parameter $w_0$ that controls the frequency range of the network. Architecturally, this approach is closely related to positional-encoding and Random Fourier feature, which also address the challenge of learning high-frequency signals (Tancik et al., 2020; Wang et al., 2021). However, SIREN requires careful tuning of $w_0$ depending on both the network architecture and the dataset (de Avila Belbute-Peres & Kolter, 2023). Moreover, the effect of network depth on performance remains poorly understood and has so far been studied mostly through empirical and observational analyses (Cai et al., 2024; Tancik et al., 2020). To the best of our knowledge, there is currently no work connecting theoretical gradient scaling with depth to the performance of such architectures.

**Neural Tangent Kernel.** The NTK framework provides a theoretical foundation for understanding the training dynamics of neural networks, and how the initialisation properties affect the learning process (Jacot et al., 2018; Li et al., 2024; Yüce et al., 2022). Some works have already focused on the frequency learning aspect of the NTK, either for the Fourier Features (Wang et al., 2021) or the SIREN architecture (de Avila Belbute-Peres & Kolter, 2023). These works have shown how the network architecture can be tailored to bypass the spectral bias. However they did not provide a full understanding of the impact of network depth on the networks properties and did not tackle the vanishing or exploding gradient impact on the learning dynamics.

**Initialisation.** Our focus on initialisation is closely related to the work of Glorot & Bengio (2010) and He et al. (2015), which introduced the now widely used Xavier and Kaiming initialisation methods, respectively. Both approaches aim to maintain stable activation and gradient distributions across layers. Xavier initialisation was developed for saturating nonlinearities such as hyperbolic tangent (Tanh), and is motivated by theoretical insights into variance preservation, though its derivation assumes an approximate linearization. Kaiming initialisation was later introduced for rectified lin-

ear units (`ReLU`), which allows for exact variance calculations. Although commonly applied to smoother activation functions such as `GeLU` or `SiLU`, its theoretical justification in those cases is only approximate. In the context of SIREN, tailored initialisations have been proposed (Sitzmann et al., 2020; de Avila Belbute-Peres & Kolter, 2023) to control the distribution of pre-activations layer by layer. However, these initialisations are only approximate and fail to offer stability guarantees for deep `SIREN` architectures, where gradient growth remains uncontrolled, as we shall see later in this work. We also note the recent work of Novello et al. (2025), which identified the same issues and proposed an *empirical* method to control a network's spectrum. However, their approach does not provide principled control of either the frequency spectrum or the network gradients, leading to significant adaptation effort for each problem.

**Edge of Chaos (EOC).** EOC is the critical initialisation regime where two key conditions are met: forward pre-activation variances remain stable, and backward gradients neither explode nor vanish. In the infinite-width mean-field limit, these properties follow from coupled recursions for layer-wise variance and inter-sample correlations under the initialisation distribution, whose fixed points determine both activation and gradient stability (Poole et al., 2016; Schoenholz et al., 2017). Yang & Schoenholz (2017) showed that placing conventional networks near the EOC improves training performance. While a vanishing-variance fixed-point initialization was previously analyzed in the EOC framework, with asymptotic Jacobian and correlation calculations for a wide range of activations (Hayou et al., 2019; Seleznova & Kutyniok, 2022; Hayou et al., 2022), including sine (Roberts et al., 2022), our work uniquely derives an explicit criticality expression for sine, a result unattainable for other activations in the sigmoid class.

## 1.3 CONTRIBUTIONS

This works brings a deeper understanding over INR initialisation for signal representation and training dynamic, with the following main contributions:

- An explicit derivation of the initialisation for the `SIREN` architecture, which allows us to have an invariant distribution of the gradients across the layers and a possibly depth-independent Fourier spectrum. This is done by calculating the fixed point for the layer-wise gradient and the network output distribution, in the limit of infinite width and infinite depth.
- The understanding of the key concepts for controlled frequency learning using $w_0$, and how the initialisation properties, through the NTK, shape the training dynamics of the network, leading to a controlled spectrum of the learned function.
- A series of experiments presented in Appendix B demonstrates the effectiveness of the proposed initialisation scheme on multi-dimensional and multi-frequency function approximation, including audio signals, image denoising, and video reconstruction on the *ERA-5* atmospheric reanalysis dataset. We further investigate the impact of this initialisation in the context of PDE solving using physics-informed neural networks.

Although our motivation comes from INRs, the proposed closed-form initialisation for sine networks at the edge of chaos is not specific to this setting. Because it stabilizes gradient propagation in deep architectures with periodic activations, it may also benefit broader applications where periodic features are desirable but have been limited by unsuitable initialisation schemes.

## 2 PRELIMINARIES

### 2.1 GENERALITIES ON IMPLICIT REPRESENTATION OF FUNCTIONS

Implicit neural representations have been introduced to find an approximation of a function $f : \Omega \mapsto \mathbb{R}^d$ from a dataset $\mathcal{D} = \{(\boldsymbol{x}_i, \boldsymbol{y}_i)_{i \in \mathbb{I}} \mid \boldsymbol{y}_i = f(\boldsymbol{x}_i)\}$. The goal is then to build a parametrized function $\Psi_\theta : \Omega \mapsto \mathbb{R}^d$. When this parametrized function is a neural network, it is commonly referred to as implicit neural representation (INR), Neural Fields (NerF), or Neural Implicit Functions.

In this work, we formally denote the involved neural network $\Psi_\theta$, which can be written as the composition of $L$ layers:

$$\Psi_\theta = h_{\theta_L} \circ \cdots \circ h_{\theta_1} \tag{1}$$

where each layer $\ell \in \{1, \ldots, L\}$ is composed of $n_\ell$ neurons, parameterized by a set of parameters $\theta_\ell = (\mathbf{W}_\ell, \mathbf{b}_\ell)$ where $\mathbf{W}_\ell \in \mathbb{R}^{n_\ell \times n_{\ell-1}}$ are the weights and $\mathbf{b}_\ell \in \mathbb{R}^{n_\ell}$ the bias, and $n_0$ denotes the input dimension of the network. Each layer also relies on an activation function $\sigma_\ell$ applied element-wise. The $\ell$-th layer is thus defined by

$$h_{\theta_\ell} = \sigma_\ell \odot \left(\mathbf{W}_\ell + \mathbf{b}_\ell\right). \tag{2}$$

For an input $\boldsymbol{x} \in \mathbb{R}^d$, the preactivation refers to

$$\mathbf{z}_\ell = \mathbf{W}_\ell \mathbf{h}_{\ell-1} + \mathbf{b}_\ell \qquad \text{where} \qquad \mathbf{h}_{\ell-1} = h_{\theta_{\ell-1}} \circ \ldots \circ h_{\theta_1}(\boldsymbol{x}). \tag{3}$$

The estimation of the parameters $\theta = \{\theta_\ell\}_{\ell \in \{1,\ldots,L\}}$ relies on the minimization of a loss $\mathcal{L}$ over a dataset $\mathcal{D} = \{(\boldsymbol{x}_i, \boldsymbol{y}_i)_{i \in \mathbb{I}}\}$:

$$\min_\theta \mathcal{L}(\theta) := \frac{1}{|\mathbb{I}|} \sum_{i \in \mathbb{I}} \|\Psi_\theta(\boldsymbol{x}_i) - \boldsymbol{y}_i\|_2^2. \tag{4}$$

The main challenges when considering INRs include selecting an appropriate architecture (i.e., parametrization and activation function), choosing a suitable initialization to insure output stability, and determining and efficient optimization strategy. In this work, we will focus on `SIREN` archithectures (described in the next section). Regarding minimization strategy, we focus on gradient-based methods, leaving alternative minimization strategies outside the scope of our study.

## 2.2 CHOICE OF THE ARCHITECTURE

This work focuses on the so called `SIREN` architecture, which stands for Sinusoidal Representation Network and introduced by Sitzmann et al. (2020). `SIREN` is a particular instance of equations 1-2 with a final linear layer:

$$\Psi_\theta(\boldsymbol{x}) = \mathbf{W}_L \sin\left(\mathbf{W}_{L-1} \sin(\ldots \sin(\mathbf{W}_1 \boldsymbol{x} + \mathbf{b}_1)) + \mathbf{b}_{L-1}\right) + \mathbf{b}_L. \tag{5}$$

This architecture enables the estimation of natural frequency decompositions in a broad range of problems while ensuring differentiability. The latter property is particularly important for PDE-related applications, such as physics-informed neural networks, where accurate derivatives are often essential (Raissi et al., 2019).

# 3 WEIGHT INITIALIZATION

In the original `SIREN` initialization (Sitzmann et al., 2020), the weights and biases were chosen as

$$\mathbf{W}_\ell \sim \begin{cases} \mathcal{U}\left(-\frac{\omega_0}{n_0}, \frac{\omega_0}{n_0}\right), & \ell = 1, \\ \mathcal{U}\left(-\frac{\sqrt{6}}{\sqrt{N}}, \frac{\sqrt{6}}{\sqrt{N}}\right), & \ell \in \{2, \ldots, L\}, \end{cases} \qquad \mathbf{b}_\ell \sim \mathcal{U}\left(-\frac{1}{\sqrt{N}}, \frac{1}{\sqrt{N}}\right), \ell \in \{1, \ldots, L\}, \tag{6}$$

where $N \equiv n_\ell$ is the number of neurons per hidden layer, assumed to be the same across all layers, $L - 1$ is the number of hidden layers, and $\mathcal{U}$ denotes the uniform distribution. $\omega_0$ is an important tunable parameter, originally chosen to be 30. It must be adjusted according to the network architecture and the Nyquist frequency of the signal to be reconstructed (de Avila Belbute-Peres & Kolter, 2023).

Sitzmann et al. (2020) argued that the pre-activation of the $\ell$-th layer, defined in equation 3, follows the distribution $\mathbf{z}_\ell \sim \mathcal{N}(0, 1)$, when the network is initialized following equation 6. In this regime, most of the signal is sufficiently small to propagates through the quasi-linear range of the sine activation function, while still preserving a meaningful nonlinear contribution. This has been emphasized as a key feature of the `SIREN` architecture. However, the initialization choice relied on approximate computations, did not provide constraints on gradients, and it has been observed that estimation quality decreases in the large-depth limit under such initialization (Cai et al., 2024). To address this, we propose the refined initialization:

$$\mathbf{W}_\ell \sim \begin{cases} \mathcal{U}\left(-\frac{\omega_0}{n_0}, \frac{\omega_0}{n_0}\right), & \ell = 1, \\ \mathcal{U}\left(-\frac{c_w}{\sqrt{N}}, \frac{c_w}{\sqrt{N}}\right), & \ell \in \{2, \ldots, L\}, \end{cases} \qquad \mathbf{b}_\ell \sim \mathcal{N}(0, c_b^2), \ell \in \{1, \ldots, L\}, \tag{7}$$

with $\mathcal{N}(0, c_b^2)$ the normal distribution of zero mean and variance $c_b^2$. This initialization introduces two parameters, $c_w$ and $c_b$, which we set by enforcing constraints on the variance of pre-activations and the rescaled layer-to-layer Jacobian:

$$\sigma_a = \lim_{\ell, N \to \infty} \sqrt{\mathrm{Var}[\mathbf{z}_\ell]} \quad \text{and} \quad \sigma_g = \lim_{\ell, N \to \infty} \sqrt{N \mathrm{Var}[\frac{\partial \mathbf{h}_{\ell+1}}{\partial \mathbf{h}_\ell}]}$$

Using explicit computations to guarantee a normalized gradient flow across the network in the mean-field limit, namely $\sigma_g = 1$, we will demonstrate in next sections that $c_b$ must lie on a curve parameterized by $c_w$:

$$c_b = \sqrt{1 - \frac{c_w^2}{3} - \frac{1}{2} \log \left( \frac{6}{c_w^2} - 1 \right)}. \tag{8}$$

We now derive two particular initialization choices along this curve. The first is the *Sitzmann-inspired* choice, obtained by enforcing $\sigma_a = 1$, which was only approximately realized in Sitzmann et al. (2020) and which we will later show does not produce the desired spectral behaviour. The second, which we adopt as our *proposed* initialization, sets $\sigma_a = 0$ and will be shown to provide much better spectral control (see Section 3.3). The corresponding parameter pairs are

$$\sigma_a = 1: \ (c_w, c_b) = \sqrt{\frac{6}{1 + e^{-2}}} \left( 1, \frac{e^{-1}}{\sqrt{3}} \right), \qquad \boldsymbol{\sigma_a = 0} \ \ \textbf{(Proposed)}: \ (c_w, c_b) = (\sqrt{3}, 0), \tag{9}$$

We illustrate the effect of these two initialization schemes on an image fitting problem in Fig. 2 and on several additional reconstruction tasks (see Appendix B). Across all depths $L$, the proposed initialization with $\sigma_a = 0$ consistently yields more stable networks than the standard SIREN (Sitzmann) architecture initialized with equation 6 and other state-of-the-art approaches. In particular, as depth increases, most competing methods exhibit gradient explosion, which manifests as spurious, noisy high-frequency artifacts in the reconstructed high-resolution images. We also find that the $\sigma_a = 1$ initialization produces slightly noisier outputs for deep networks than the $\sigma_a = 0$ scheme, a behaviour explained in Section 3.3 and motivating our preference for the proposed initialization.

### 3.1 Pre-activation distribution

In the following, we derive the exact form of the pre-activation distribution in the limit of infinitely wide and deep neural networks, explicitly accounting for the influence of the bias term, which turns out to be crucial. More precisely, we show that, for any initialization in the parameter space $(c_w, c_b)$, the pre-activation distribution converges to a fixed point. The proof is provided in Appendix A.1.

**Theorem 3.1** (Pre-activation distribution of SIREN). *Considering SIREN network described in equation 5 where, for some $c_w, c_b \in \mathbb{R}^+$, and for every layer $\ell \in \{2, \ldots, L\}$, the weight matrix $\mathbf{W}_\ell$ is initialized as a random matrix sampled from $\mathcal{U}(-c_w/\sqrt{N}, c_w/\sqrt{N})$, $\mathbf{W}_1$ is sampled from $\mathcal{U}(-w_0/n_0, w_0/n_0)$, the bias $\mathbf{b}_\ell$ is initialized as a random vector sampled from $\mathcal{N}(0, c_b^2)$. Let $(\mathbf{z}_\ell)_{\ell \in \{1 \ldots, L\}}$ the pre-activation sequence defined in equation 3 and relying on an input $x \in \mathbb{R}^{n_0}$. Then, in the limits $N, L \to \infty$, the pre-activation sequence $(\mathbf{z}_\ell)_{\ell \in \mathbb{N}}$ converges in distribution to $\mathcal{N}(0, \sigma_a^2)$ with*

$$\sigma_a^2 = c_b^2 + \frac{c_w^2}{6} + \frac{1}{2} \mathcal{W}_0 \left( -\frac{c_w^2}{3} e^{-\frac{c_w^2}{3} - 2c_b^2} \right), \tag{10}$$

*where $\mathcal{W}_0$ is the principal real branch of the Lambert function. The sequence associated to the variance of the pre-activation $\big( \mathrm{Var}(\mathbf{z}_\ell) \big)_{\ell \in \mathbb{N}}$ converges to a fixed point $\sigma_a$, which is exponentially attractive for all values of $c_w \neq \sqrt{3}$. For $c_w = \sqrt{3}$ the convergence will be of rate $\mathcal{O}(\frac{1}{\ell})$.*

**Remark 3.1.** *While the bias distribution is different in our initialization and in the original SIREN scheme, the choice $c_w = \sqrt{6}$ for the weight initialization can be recovered as a special case of equation 10 by imposing $\sigma_a = 1$, assuming $c_b = 0$, and by neglecting the correction term introduced by the Lambert function. Using the expansion $\mathcal{W}_0(x) = x + \mathcal{O}(x^2)$, this correction term can be estimated as $\sim e^{-2}$, which is small but not negligible[1]. Accounting for this correction term enables more precise control over the pre-activation variance $\sigma_a$.*

---

[1] A more precise estimate of this correction term can be obtained using equation 29, to be derived later.

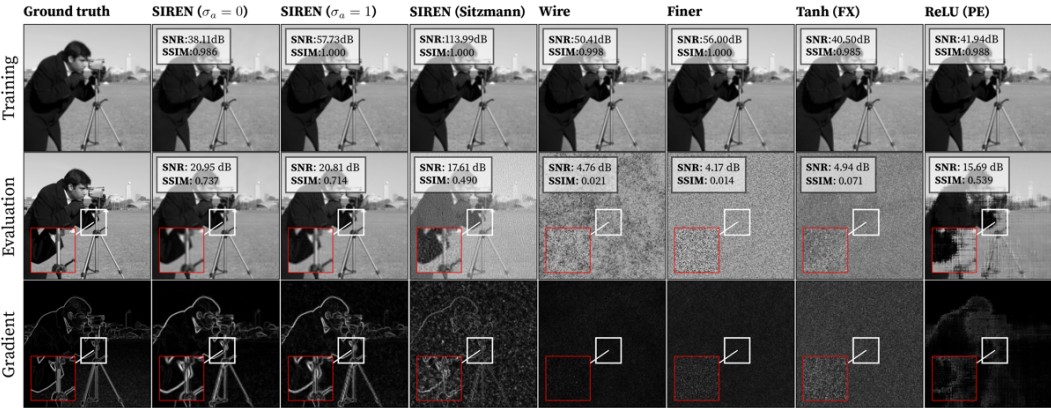

Figure 2: Comparison of several INR architectures and initializations on an image-fitting problem using an $L = 10$ hidden-layer neural network of width $N = 256$. We train the model on a set $(\boldsymbol{x}_i, y_i)_{i \in \mathbb{I}}$ where $\boldsymbol{x}_i$ is a location taken on a $|\mathbb{I}| = 128 \times 128$ uniformly spaced grid on $\Omega = [-1, 1]^2$ and $y_i$ is the associated image value at this location. The top row shows the fitted $128 \times 128$ image. The middle row shows the estimation on an augmented resolution ($512 \times 512$) to assess the model's generalization and the last row provides a zoom on part of the image. In all case, we use ADAM optimizer with learning rate $10^{-4}$ for 10000 epochs. The state-of-the-art architecture considered in this experiment are: `SIREN` (see (Sitzmann et al., 2020)), `FINER` (see (Liu et al., 2024)), `WIRE` (see (Saragadam et al., 2022)), `Tanh(FX)` with Fourier features and Xavier initialization (see (Tancik et al., 2020)), and the traditional `ReLU` with Positional Encoding (see (Nair & Hinton, 2010)). We used for the `SIREN` based architectures the previously discussed schemes. We observe that the proposed strategies (`SIREN` ($\sigma_a = 0$ and $\sigma_a = 1$) lead to significant improvement in the model estimation with respect to other methods. For instance, it preserves sharp features compared to other SOTA method such as `Wire`, `Finer`, that yields extremely poor results for deep neural networks.

**Remark 3.2.** *As stated in Theorem 3.1, the pre-activation variance converges exponentially fast to $\sigma_a$ as the depth $L$ increases whenever $c_w \neq \sqrt{3}$. In that case, even relatively shallow networks already have pre-activations that are effectively Gaussian with variance very close to the fixed point $\sigma_a$. When $c_w = \sqrt{3}$, this convergence becomes much slower. For our proposed choice $\sigma_a = 0$, this means that the pre-activation variance decays toward zero only gradually with depth.*

Deriving the fixed points of the pre-activation distribution is a necessary first step toward characterizing the layer-wise gradient distribution and for establishing the optimal initialization value for $c_w$ and $c_b$, which we discuss in the next subsection.

## 3.2 GRADIENT DISTRIBUTION AND STABILITY

The distribution of Jacobian entries is another important property of neural networks that must be carefully controlled during initialization to avoid gradient vanishing (He et al., 2015; Yang & Schoenholz, 2017). In this work, we show that a tractable derivation is possible for the sine activation function. This result is described in Theorem 3.2. Combined with Theorem 3.1 it will enable us to propose a principled initialization strategy provided in Proposition 3.1.

**Theorem 3.2** (Jacobian distribution of `SIREN`). *Let $\mathbf{J}_\ell = \partial\mathbf{h}_\ell / \partial\mathbf{h}_{\ell-1}$ denote the Jacobian of the $\ell$-th layer. Considering `SIREN` network described in equation 5, we have*

$$\mathbf{J}_\ell = \mathrm{diag}(\cos(\mathbf{z}_\ell)) \, \mathbf{W}_\ell.$$

*Under the same assumptions as Theorem 3.1, and maintaining the limit of large $N$, each entry of $\mathbf{J}_\ell$ has zero mean and a variance $\widetilde{\sigma}_\ell^2$, such that the sequence $(N\widetilde{\sigma}_\ell^2)_{\ell \in \mathbb{N}}$ converges to*

$$\lim_{\ell, N \to \infty} (N\widetilde{\sigma}_\ell^2) = \sigma_g = \frac{c_w^2}{6}\big(1 + e^{-2\sigma_a^2}\big). \tag{11}$$

For a given network, with input $\boldsymbol{x}$ and output $\Psi_\theta(\boldsymbol{x})$, Theorem 3.2 can be used to analyze the scaling behavior of gradients with respect to both the network parameters $\theta$ and the input coordinates $\boldsymbol{x}$. We

denote by $\partial_{\theta_\ell}\Psi$ the gradient of the network output with respect to the parameters $\theta_\ell$ of layer $\ell$, and by $\partial_{\boldsymbol{x}}\Psi$ the gradient with respect to the input $\boldsymbol{x}$. By applying the chain rule, we have :

$$\frac{\partial\Psi_\theta(\boldsymbol{x})}{\partial\theta_\ell} = \frac{\partial\Psi_\theta}{\partial\mathbf{h}_{L-1}}\cdots\frac{\partial\mathbf{h}_{\ell+1}}{\partial\mathbf{h}_\ell}\frac{\partial\mathbf{h}_\ell(\boldsymbol{x})}{\partial\theta_\ell}, \quad \frac{\partial\Psi_\theta(\boldsymbol{x})}{\partial\boldsymbol{x}} = \frac{\partial\Psi_\theta(\boldsymbol{x})}{\partial\mathbf{h}_{L-1}}\cdots\frac{\partial\mathbf{h}_2}{\partial\mathbf{h}_1}\frac{\partial\mathbf{h}_1(\boldsymbol{x})}{\partial\boldsymbol{x}}. \quad (12)$$

These relations can be used to obtained scaling of the gradients variances with the network depth and width (see Appendix A.4 for a derivation):

$$\mathrm{Var}(\partial_{\theta_\ell}\Psi_\theta(\boldsymbol{x})) \propto N^{-1}\left(\sigma_g^2\right)^{L-\ell-1} \quad\text{and}\quad \mathrm{Var}(\partial_{\boldsymbol{x}}\Psi_\theta(\boldsymbol{x})) \propto \omega_0^2\left(\sigma_g^2\right)^{L-2}. \quad (13)$$

From equation 13, we see that gradients in parameter space vanish or explode exponentially with network depth $L$, unless the scaling factor $\sigma_g^2$ is close to 1. To conclude the analysis of the statistical properties of SIREN networks and derive the initialization schemes provided in equations 7-9, we identify the values of $c_w$ and $c_b$ allowing to control the scaling of gradients i.e. $\sigma_g = 1$.

**Proposition 3.1.** *Under the same assumptions as in Theorem 3.1, setting $\sigma_g = 1$ leads to the weight–bias variance curve $c_b(c_w)$ defined in equation 8. Furthermore, choosing $\sigma_a = 0$ (our proposed initialization) or $\sigma_a = 1$ determines a specific pair $(c_w, c_b)$ given in equation 9.*

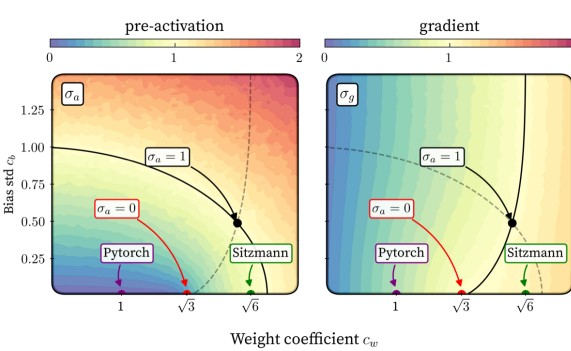

Figure 3: Experimental standard deviation of the pre-activation distribution (left) and of the layer-wise Jacobian entries distribution (right), as a function of the parameters $(c_w, c_b)$. The plain and dashed black lines indicate the theoretical predictions for $\sigma_a = 1$ and $\sigma_g = 1$, following Theorems 3.1 and 3.2, respectively. The black and red dots indicates the initialization provided in Proposition 3.1, the Pytorch dots corresponds to the default weight and bias initialization, and the green dots to the Sitzmann initialization.

The proof is given in Appendix A.3. We verified the validity of this theoretical analysis, involving careful calculations of the Jacobian and pre-activation distributions, through numerical experiments displayed in figure 3. These experiments were done 20 times using a SIREN neural network of width $N = 256$ of depth $L = 10$, with input dimension $n_0 = 1$, and output dimension $n_d = 1$, $w_0 = 1$, and following the initialization scheme in equations 7-9. The neural network is then evaluated using $|\mathbb{I}| = 500$ input points $\boldsymbol{x}_i$ uniformly spaced between $[-1, 1]$ to obtain the studied distributions.

In the next section, we explain why choosing $\sigma_a = 0$ rather than $\sigma_a = 1$ provides better control over the network's frequency spectrum.

## 3.3 FOURIER SPECTRUM AND ALIASING

The need to constrain the Fourier spectrum of sinusoidal neural networks to prevent high-frequency aliasing was noted in (Yüce et al., 2022), and a closed-form expression for the spectrum of sine-based networks was later derived in (Novello et al., 2025, Thm. 3), showing that each additional layer redistributes energy across Fourier modes. Since composing sine activations inherently broadens the spectrum with depth, controlling this growth requires either limiting the depth or enforcing $\sigma_a = 0$. In the latter case, deep layers are almost linear, because for $\mathbf{z}_\ell \sim \mathcal{N}(0, \sigma_a^2)$ we have $\sin(\mathbf{z}_\ell) \approx \mathbf{z}_\ell$ as $\sigma_a \to 0$. Empirically, our initialization with $\sigma_a = 0$ indeed suppresses the emergence of higher frequencies: as shown in Fig. 4, spectral broadening with depth is strongly reduced, and most of the energy remains confined below $w_0$, yielding a meaningful, depth-independent cutoff around $w_0$. The slow decay of $\sigma_\ell$ toward zero described in Theorem 3.1 appears to compensate the nonlinearities just enough to avoid both explosion and collapse of the spectrum, even in very deep networks, a behaviour that remains unexplained and calls for further investigation.

In contrast, for $\sigma_a = 1$, and even more so under the Sitzmann initialization, the spectrum clearly broadens with depth, and substantial energy appears beyond $w_0$. This excess energy is exactly what

causes aliasing when the network input is discretized. For the PyTorch initialization, the opposite behavior occurs: the spectrum collapses rapidly with depth, reflecting a vanishing-signal regime caused by unnormalized gradients. Overall, this analysis supports our proposed initialization, which constrains $\sigma_a = 0$ and motivates choosing $w_0$ as the Nyquist frequency for sampled inputs. This ensures that the network can represent all frequencies present in the data while avoiding aliasing in the early stages of training.

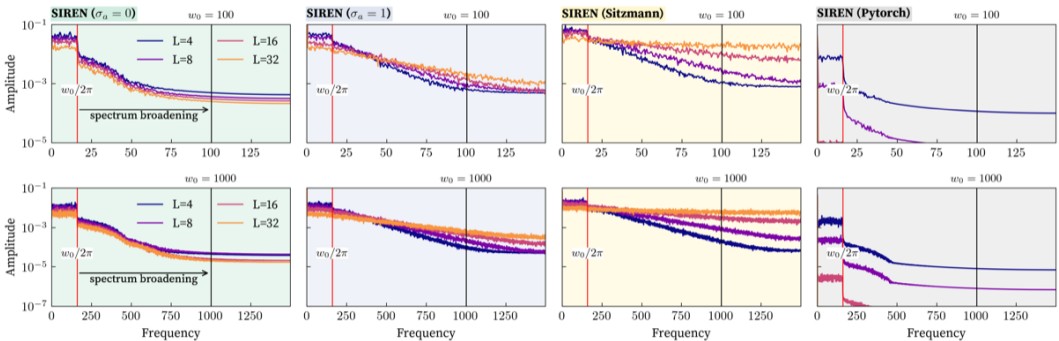

Figure 4: One-dimensional Fourier spectra of $\Psi_\theta$ for multiple depths $L \in \{4, 8, 16, 32\}$, driving frequencies $w_0 \in \{100, 1000\}$ (rows), and initialization schemes (columns). Each curve shows the magnitude of the discrete Fourier transform of $\Psi_\theta$ evaluated on an equispaced grid; colors encode the depth $L$. The red vertical line marks $w_0/2\pi$ which corresponds to the input frequency encoded by the first layers and the black vertical line marks $w_0$. The colored backgrounds group the different initializations (from left to right: proposed SIREN with $\sigma_a = 0$, SIREN with $\sigma_a = 1$, the initialization of (Sitzmann et al., 2020), and the default PyTorch initialization).

# 4 SCALING OF THE NEURAL TANGENT KERNEL WITH DEPTH AND SIMPLIFIED LEARNING DYNAMICS

**The Neural Tangent Kernel (NTK)** framework is a linearized description of the training dynamics around initialization, allowing one to study how the network evolves in the early phase of training (Jacot et al., 2018). When training neural networks, we typically use gradient descent to minimize the loss function, with updates $\theta_{t+1} = \theta_t - dt\nabla_\theta \mathcal{L}(\theta_t)$, where $dt$ is the learning rate and $\theta_t$ the parameter vector at iteration $t$.

To simplify we restrict ourselves to a scalar output neural network (i.e., $d = 1$). Then, we have for the mean-squared error loss $\mathcal{L}(\theta) = \sum_{i \in \mathbb{I}} \|\Psi_\theta(\boldsymbol{x}_i) - y_i\|^2/|\mathbb{I}|$, and in the continuous-time limit $dt \to 0$, the residuals $u(\boldsymbol{x}_i, t) = \Psi_{\theta_t}(\boldsymbol{x}_i) - y_i$ satisfy

$$\frac{d\boldsymbol{u}(t)}{dt} = \mathbf{K}_{\theta_t}\boldsymbol{u}(t), \quad \mathbf{K}_{\theta_t,i,j} = \nabla_\theta \Psi_{\theta_t}(\boldsymbol{x}_i) \cdot \nabla_\theta \Psi_{\theta_t}(\boldsymbol{x}_j), \tag{14}$$

where $\boldsymbol{u}(t) = (u(\boldsymbol{x}_1, t), \ldots, u(\boldsymbol{x}_{|\mathbb{I}|}, t))$ and $\mathbf{K}_{\theta_t}$ is the NTK matrix. Assuming the NTK remains constant during training ($\mathbf{K}_{\theta_t} \equiv \mathbf{K}_{\theta_0}$), the residuals evolve as

$$\boldsymbol{u}(t) = \exp(-t\mathbf{K}_{\theta_0})\boldsymbol{u}(0) = \sum_{i=1}^{|\mathbb{I}|} e^{-t\lambda_i}\langle\boldsymbol{u}(0), \boldsymbol{v}_i\rangle\boldsymbol{v}_i, \tag{15}$$

where $(\lambda_i, \boldsymbol{v}_i)$ are the eigenpairs of the initialized NTK $\mathbf{K}_{\theta_0}$, ordered so that $\lambda_1 \geq \cdots \geq \lambda_{|\mathbb{I}|} > 0$, and $\langle\cdot, \cdot\rangle$ the Euclidean scalar product. Thus, the early training dynamics is fully determined by the spectral properties of the NTK at initialization.

**Frequency bias in the NTK framework.** Equation 15 shows that modes associated with large eigenvalues decay quickly, while those with small eigenvalues decay slowly, with characteristic timescale $1/\lambda_i$. As illustrated in Fig. 5 for the 1D case, and as observed in related settings (see e.g. (Wang et al., 2021)), the leading eigenmodes (small $i$) of the NTK can be identified with low-frequency Fourier modes, whereas higher-frequency components (large $i$) correspond to smaller

eigenvalues $\lambda_i$. Figure 5 provides an overview of this behavior. This illustrates the spectral bias of neural networks in the lazy training regime (i.e., nearly constant NTK) and emphasizes the importance of controlling the spectrum $\{\lambda_i\}_{i=1}^{|\mathbb{I}|}$ to accurately capture all relevant target frequencies. A more detailed study of the overlap between NTK and Fourier modes, for different initialisation schemes, is presented in Appendix B.2.2.

**First 6 eigenvectors of the NTK of a SIREN**

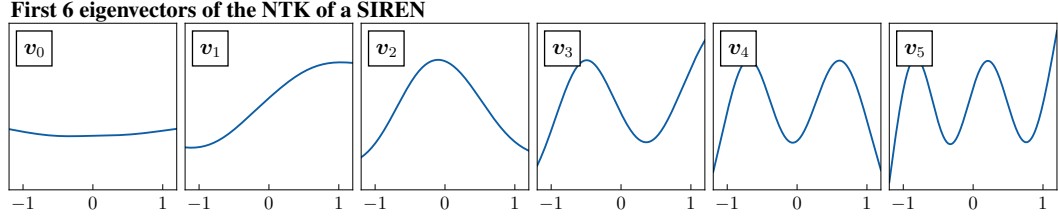

Figure 5: The first six eigenvectors $v_0, \ldots, v_5$ of the NTK matrix $\mathbf{K}_{\theta_0}$, ordered by decreasing eigenvalue $\lambda_0 > \lambda_1 > \cdots > \lambda_5$. The NTK matrix was computed numerically on a uniform grid of $|\mathbb{I}| = 500$ points over the interval $\Omega = [-1, 1]$ using a SIREN network of width $N = 512$ and of depth $L = 8$ and using $\omega_0 = 1$. The eigenvectors exhibit increasingly oscillatory behavior as the mode index grows, consistent with their interpretation as Fourier-like modes. This observation confirms the spectral structure predicted by our analysis and highlights the tendency of the NTK to prioritize low-frequency components associated with larger eigenvalues.

**Empirical scaling of NTK eigenvalues and network gradients.** To highlight the importance of initialization in the large depth limit, we conducted an experiment comparing the original SIREN initialization (cf. equation 6), the new ones (cf. equations 7-9), and the Pytorch one. We varied the depth $L$ while fixing $N = 256$, $|\mathbb{I}| = 200$, and $\omega_0 = 1$. In figure 6, we plot the normalized NTK trace (mean eigenvalue) expressed as $\mathrm{Tr}(\mathbf{K}_{\theta_0})/(|\mathbb{I}|N)$, together with the gradient norm $\|\partial_x \Psi_{\theta_0}\|$ as functions of network depth. We use the NTK trace as a computationally convenient proxy for the typical eigenvalue behavior as depth increases. With the original SIREN initialization, we observe exponential growth of both the NTK eigenvalues and the input gradients. In this case, increasing depth accelerates training but also causes gradient explosion in input space. This corresponds to spurious high-frequency components absent from the target signal, which degrade generalization, here understood as smooth interpolation between data points. With PyTorch initialization, the NTK eigenvalues decrease until reaching a plateau, while the gradient in input coordinate space vanishes. By contrast, with our new initialisations, the NTK eigenvalues increases linearly with depth while the gradients remain constant. Consequently, the effective learning rate increases with depth $L$, while the input-space gradients stay normalized. These behaviors are confirmed in practical settings, such as the image-fitting task shown in figure 2, and in additional experiments presented in Appendix B.

**Interpretation of the scalings.** The scaling of gradients with $\sigma_g^L$ is expected from section 3.2, with $\sigma_g \approx \sqrt{1.2}$ for SIREN, $\sigma_g = 1$ for our proposed initialization, and $\sigma_g = \sqrt{1/3}$ for PyTorch initialization. Similarly, it is possible to explain the NTK eigenvalue scaling. We note first that diagonal element of the NTK matrix are $\mathbf{K}_{\theta_0,i,i} = |\nabla_\theta \Psi_{\theta_0}(x_i)|^2$. From this and the zero mean property of every gradient distribution, we relate the average eigenvalue of the NTK denoted $\bar{\lambda}$ to the variance of gradients in parameter space:

$$\bar{\lambda} = \frac{1}{|\mathbb{I}|} \mathrm{Tr}(\mathbf{K}_{\theta_0}) = N^2 \sum_{\ell=1}^{L} \mathrm{Var}\left[\nabla_{\mathrm{W}_\ell} \Psi_{\theta_0}(x_i)\right] + N \sum_{\ell=1}^{L} \mathrm{Var}\left[\nabla_{\mathrm{b}_\ell} \Psi_{\theta_0}(x_i)\right],$$ (16)

where $\mathrm{W}_\ell, \mathrm{b}_\ell$ are respectively a weight and a bias of the $\ell$-th layer. The sum involving weights parameters being dominant, we neglect the sum on bias terms in the following. When $\sigma_g^2 \neq 1$, using equation 12, we obtain a geometric sum, leading to

$$\frac{1}{|\mathbb{I}|N} \mathrm{Tr}(\mathbf{K}_{\theta_0}) \propto \frac{(\sigma_g^2)^{L+1} - 1}{\sigma_g^2 - 1}.$$ (17)

If $\sigma_g > 1$ (`SIREN` original), then $\bar{\lambda} \propto \sigma_g^{2L}$ and the NTK explodes exponentially with depth $L$. This exponential scaling for the NTK eigenvalues without proper initialization was observed experimentally in (de Avila Belbute-Peres & Kolter, 2023), yet without precise discussion on the causes and the effect of such behavior, since their focus was on the choice of $\omega_0$ rather than on weight and bias initialization.

If $\sigma_g < 1$ (`SIREN` PyTorch), NTK eigenvalues become independent from the depth $L$ in the large depth limit, yielding slow convergence, together with vanishing gradients.

If $\sigma_g = 1$ (`SIREN` $\sigma_a = 0, 1$), equation 17 does not apply. Each term of the sum on weight parameters in equation 16 gives the same contribution, leading to $\bar{\lambda} \propto L$, which is consistent with the results plotted figure 6 for the $\sigma_a = 1$ initialization, for $\sigma_a = 0$ it seems that the NTK eigenvalues are converging to a fix distribution, and we attribute that to finite size effect of our initialization, indeed the convergence is really slow towards $\sigma_a = 0$, which seems to compensate the NTK eigenvalues growth with depth, for finite depth networks.

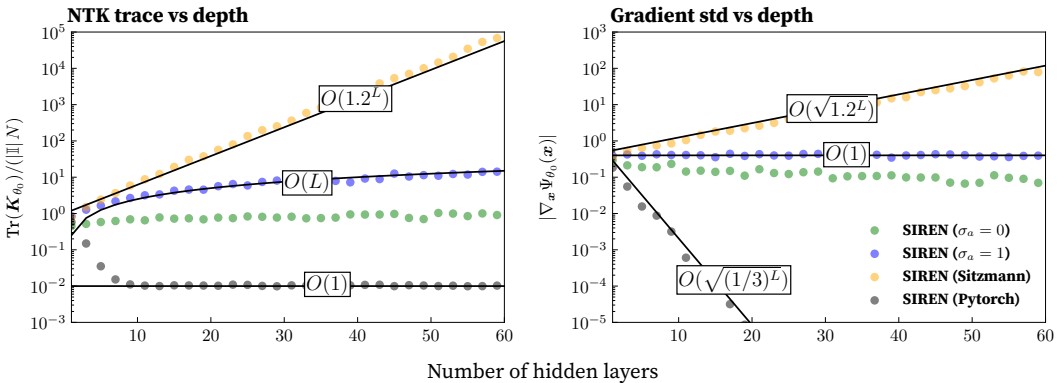

Figure 6: The left plot stands for the scaling of the mean eigenvalue of the NTK matrix over the number of layer. The right plot stands for the scaling of the gradient of the network (in input coordinate space) with the number of layers. The experimental setup and hyper-parameters are the same as in figure 5, except for the network depth which varies here.

## 5 DISCUSSION, CONCLUSION, PERSPECTIVES

We proposed a new initialization scheme for sinusoidal neural networks that prevents gradient explosion and vanishing, and presented various applications, from noisy image fitting, video, and audio reconstruction (Appendix B). The parametrization is derived analytically by examining the variances of pre-activations and layer-to-layer Jacobians in the limit of infinitely wide and deep networks. This approach removes the need for architectural tricks such as skip connections or empirical hyperparameter tuning to stabilize deep models. By analyzing both the neural tangent kernel and input-space gradients, we showed that this initialization enables deep networks to train with learning rates that scale linearly with depth, while suppressing spurious noise above the Nyquist frequency in implicit neural representations. Whereas prior work motivated the use of sine activations by noting that derivatives of SIRENs remain well-behaved, our study goes further by providing a deeper theoretical analysis. We demonstrate that sinusoidal architectures not only preserve these desirable properties but also admit stronger theoretical justification. A key take-away is that fixing the Jacobian variance ($\sigma_g = 1$) is essential to control gradients, whereas setting the targeted fixed point pre-activation variance ($\sigma_a = 0$) gives direct control over the network spectrum at initialization.

Future work could extend the approach to more complex losses, including physics-informed settings, and to potential applications to atmospheric and oceanic field reconstruction based on INRs (Johnson et al., 2022; Mostajeran et al., 2025). Furthermore, our study focuses solely on controlling the variance of the weights at initialization. One could broaden this perspective by considering additional structural properties of the network such as the distribution of singular values of the layer Jacobians (presented in Appendix B.1).

## REPRODUCIBILITY

**Code Implementation.** All source code used in our experiments is provided in the supplementary material, including implementations of the architectures used for comparison.

**Models and Architectures.** Details on the choice of activation functions are given in the main text. Initialization methods and architectural specifications for each model are described within the corresponding experimental sections.

**Experiments.** Each experiment is reported with its hyperparameters (e.g., learning rate, optimizer, number of epochs) in the relevant sections or figures. All experiments were run with fixed random seeds to ensure exact reproducibility of the reported results.

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

# A  MATHEMATICAL APPENDIX

## A.1  INPUT DISTRIBUTION

**Theorem** (Restatement of Theorem 3.1). *Considering SIREN network described in equation 5 where, for some $c_w, c_b \in \mathbb{R}^+$, and for every layer $\ell \in \{2, \ldots, L\}$, the weight matrix $\mathbf{W}_\ell$ is initialized as a random matrix sampled from $\mathcal{U}(-c_w/\sqrt{N}, c_w/\sqrt{N})$, and the bias $\mathbf{b}_\ell$ is initialized as a random vector sampled from $\mathcal{N}(0, c_b^2)$. Let $(\mathbf{z}_\ell)_{\ell \in \{1 \ldots, L\}}$ the preactivation sequence defined in equation 3 and relying on an input $\boldsymbol{x} \in \mathbb{R}^{n_0}$. Then, in the limit of large $N$, the preactivation sequence $(\mathbf{z}_\ell)_{\ell \in \mathbb{N}}$ converges in distribution to $\mathcal{N}(0, \sigma_a^2)$ where*

$$\sigma_a^2 = c_b^2 + \frac{c_w^2}{6} + \frac{1}{2}\mathcal{W}_0\left(-\frac{c_w^2}{3}e^{-\frac{c_w^2}{3}-2c_b^2}\right) \tag{18}$$

*with $\mathcal{W}_0$ is the principal real branch of the Lambert function. Additionally, the sequence associated to the variance of the preactivation $\left(\mathrm{Var}(\mathbf{z}_\ell)\right)_{\ell \in \mathbb{N}}$ converges to a fixed point $\sigma_a$, which is exponentially attractive for all values of $c_w \neq \sqrt{3}$.*

*Proof.* The proof can be split in three steps: (i) prove that the sequence of preactivations follows a Gaussian distribution (cf. Lemma A.1), (ii) give an expression of the variance of the output of a sin activation when the input follows a zero-mean Gaussian distribution of s.t.d. $\sigma_a$ (cf. Lemma A.2), (iii) provides the expression of the variance of each element of the preactivation sequence using the result in (ii) and proves its convergence to a fixed point $\sigma_a$ (cf. Lemma A.3).

**Lemma A.1.** *Considering SIREN network described in equation 5 where, for some $c_w, c_b \in \mathbb{R}^+$, and for every layer $\ell \in \{2, \ldots, L\}$, the weight matrix $\mathbf{W}_\ell$ is initialized as a random matrix sampled from $\mathcal{U}(-c_w/\sqrt{N}, c_w/\sqrt{N})$, $\mathbf{W}_1$ is sampled from $\mathcal{U}(-w_0/n_0, w_0/n_0)$, and the bias $\mathbf{b}_\ell$ is initialized as a random vector sampled from $\mathcal{N}(0, c_b^2)$. Let $(\mathbf{z}_\ell)_{\ell \in \{1 \ldots, L\}}$ the preactivation sequence defined in equation 3 and relying on an input $\boldsymbol{x} \in \mathbb{R}^{n_0}$. Then, in the limit of large $N$, each element of the preactivation sequence $(\mathbf{z}_\ell)_{\ell \in \mathbb{N}}$ is distributed according to a zero-mean Gaussian distribution.*

*Proof.* We recall that for the first layer, $\mathbf{h}_0 = \boldsymbol{x}$ and, for every $\ell \in \{1, \ldots, L\}$,

$$\mathbf{h}_\ell = \sin\left(\mathbf{W}_\ell \mathbf{h}_{\ell-1} + \mathbf{b}_\ell\right).$$

Since the sine activation is an odd function, it preserves the zero-mean property of any distribution: if $\mathbf{z}_\ell = \mathbf{W}_\ell \mathbf{h}_{\ell-1} + \mathbf{b}_\ell$ has zero mean, then $\mathbf{h}_\ell$ will also have zero mean. This property propagates layer by layer.

As $\mathbf{W}_1$ and $\mathbf{b}_1$ are assumed to have zero mean (by definition, cf. equation 7) and $\boldsymbol{x}$ is a deterministic vector, it ensures that the first-layer pre-activation has zero-mean. Moreover, as $\mathbf{W}_\ell$ and $\mathbf{b}_\ell$ are assumed to have zero mean the zero-mean property holds for all subsequent pre-activations $\mathbf{z}_\ell$ and $\mathbf{h}_\ell$.

Second, we prove that the preactivation sequence is distributed according to a Gaussian. We first rewrite each element of the preactivation sequence as

$$\mathbf{z}_{\ell,i} = \sum_{j=1}^{N} \mathbf{W}_{\ell,i,j}\mathbf{h}_{\ell-1,j} + \mathbf{b}_{\ell,i}\,. \tag{19}$$

As a sum of two Gaussian stays Gaussian and because $\mathbf{b}_\ell$ is assumed to be Gaussian with a standard deviation $\sigma_b$, the main purpose here is then to prove that $\sum_{j=1}^{N} \mathbf{W}_{\ell,i,j}\mathbf{h}_{\ell-1,j}$ follow a Gaussian distribution.

Thanks to the Central Limit Theorem, whatever is the distribution of $\mathbf{h}_{\ell-1,j}$, the term $\sum_{j=1}^{N} \mathbf{W}_{\ell,i,j}\mathbf{h}_{\ell-1,j}$ converges in distribution to a Gaussian distribution in the limit of large $N$. Since the bias is also normally sampled, each component $\mathbf{z}_{\ell,i}$ follows a gaussian distribution in the same large $N$ limit, with zero mean and a variance denoted $\sigma_a^2$.

To compute this variance, let us first compute the variance of each summand denoted $\sigma^2_{\ell,i,j}$, given by the product of two independent random variables with zero mean, namely $W_{\ell,i,j}$ and $h_{\ell-1,j}$,

$$\sigma^2_{\ell,i,j} = \text{Var}\left[W_{\ell,i,j}\right]\text{Var}\left[h_{\ell-1,j}\right], . \tag{20}$$

Since $W_{\ell,i,j}$ is uniformly distributed on $[-c_w/\sqrt{N}, c_w/\sqrt{N}]$, we have:

$$\text{Var}\left[W_{\ell,i,j}\right] = \frac{c_w^2}{3N}. \tag{21}$$

While the variance of $h_{\ell-1,j}$ is still unknown, we can express it from the knowledge of $\mathbf{z}_{\ell-1}$, leading to

$$\sigma^2_{\ell,i,j} = \frac{c_w^2}{3N}\text{Var}\left[\sin(z_{\ell-1,j})\right]. \tag{22}$$

whose expression of $\text{Var}\left[\sin(z_{\ell-1,j})\right]$ will be provided later.

As the bias variance follows a Gaussian distribution as described in equation 7, the variance of all the elements of the preactivation $\mathbf{z}_\ell$ is

$$\sigma^2_\ell = \frac{c_w^2}{3}\text{Var}\left[\sin(\mathbf{z}_{\ell-1})\right] + c_b^2. \tag{23}$$

$\square$

**Lemma A.2.** *Let $z$ be a normally distributed random variable and zero mean $z \sim \mathcal{N}(0, \sigma^2)$. Then we have :*

$$\text{Var}\left[\sin(z)\right] = \frac{1}{2}\left(1 - e^{2\sigma^2}\right). \tag{24}$$

*Proof of Lemma A.2.* The proof combined the properties of the Gaussian distribution with the fact that the sine function is an odd function. We have:

$$\text{Var}\left[\sin(z)\right] = \mathbb{E}\left[\sin^2(z)\right] - \mathbb{E}\left[\sin(z)\right]^2$$

Since $\sin$ is odd and since the expectation of $z$ is zero, we have $\mathbb{E}\left[\sin(z)\right] = 0$. In addition, using $\sin^2(z) = (1 - \cos(2z))/2$, we obtain

$$\mathbb{E}\left[\sin^2(z)\right] = \frac{1}{2} - \frac{1}{2}\mathbb{E}\left[\cos(2z)\right].$$

The characteristic function of the Gaussian distribution with zero mean and variance $\sigma_a$ is given by:

$$g_z(t) = \mathbb{E}(e^{itz}) = e^{-\frac{1}{2}t^2\sigma^2}.$$

Now we notice that

$$\mathbb{E}\left[\cos(2z)\right] = \mathbb{E}\left[\Re\left[e^{i2z}\right]\right] = \Re\left[g_z(2)\right] = e^{-2\sigma_a^2}.$$

The first equality uses the linearity of the mean. This leads to the final result:

$$\text{Var}\left[\sin(z)\right] = \frac{1}{2}\left(1 - e^{-2\sigma^2}\right).$$

$\square$

**Lemma A.3.** *Considering SIREN network described in equation 5 where, for some $c_w, c_b \in \mathbb{R}^+$, and for every layer $\ell \in \{1, \ldots, L\}$, the weight matrix $\mathbf{W}_\ell$ is initialized as a random matrix sampled from $\mathcal{U}(-c_w/\sqrt{N}, c_w/\sqrt{N})$, and the bias $\mathbf{b}_\ell$ is initialized as a random vector sampled from $\mathcal{N}(0, c_b^2)$. Let $\mathbf{x} \in \mathbb{R}^{n_0}$. Then, in the limit of large $N$, the preactivation sequence $(\mathbf{z}_\ell)_{\ell \in \{1\ldots,L\}}$ defined in equation 3 is distributed according to a Gaussian distribution with zero-mean and, for every $\ell$, a variance*

$$\sigma^2_\ell = \frac{c_w^2}{6}\left(1 - e^{-2\sigma^2_{\ell-1}}\right) + c_b^2$$

*Moreover, the sequence $(\sigma^2_\ell)_{\ell \in \mathbb{N}}$ converges to*

$$\sigma_a^2 = c_b^2 + \frac{c_w^2}{6} + \frac{1}{2}\mathcal{W}_{0,-1}\left(-\frac{c_w^2}{3}e^{-\frac{c_w^2}{3}-2c_b^2}\right),$$

*with $\mathcal{W}_{0,-1}$ the two real branches of the Lambert W function. And for $c_w \neq \sqrt{3}$, this convergence is exponentially fast.*

*Proof of Lemma A.3.*

**Fixed Point Value :** Combining equation 23 and equation A.3, the variance of the pre-activation at layer $\ell$ is

$$\sigma_\ell^2 = \frac{c_w^2}{6}\left(1 - e^{-2\sigma_{\ell-1}^2}\right) + c_b^2$$

To characterize the fixed point of the sequence $(\sigma_\ell^2)_{\ell \in \mathbb{N}}$, we define a function $f$ as

$$f(x) = \frac{c_w^2}{6}\left(1 - e^{-2x}\right) + c_b^2. \tag{25}$$

The fixed point of this function is given by the solution of the equation $f(x) = x$. Rearranging the different term gives:

$$\frac{c_w^2}{6} + c_b^2 - x = \frac{c_w^2}{6}e^{-2x}. \tag{26}$$

Using $y = \frac{c_w^2}{6} + c_b^2 - x$ yields

$$ye^{-2y} = \frac{c_w^2}{6}e^{-2(\frac{c_w^2}{6}+c_b^2)}.$$

Then, using the definition of the real valued Lambert W function, we get

$$y = -\frac{1}{2}\mathcal{W}_k\left(-\frac{c_w^2}{3}e^{-2(\frac{c_w^2}{6}+c_b^2)}\right), \quad \text{where} \quad k \in \{-1, 0\}.$$

The $\mathcal{W}_0$ branch is called the principal branch and is defined on $(-e^{-1}, +\infty)$. The $\mathcal{W}_{-1}$ branch is defined for $(-e^{-1}, 0)$. To obtain a positive variance, the branch to consider is $\mathcal{W}_0$, as illustrated numerically in figure 7.

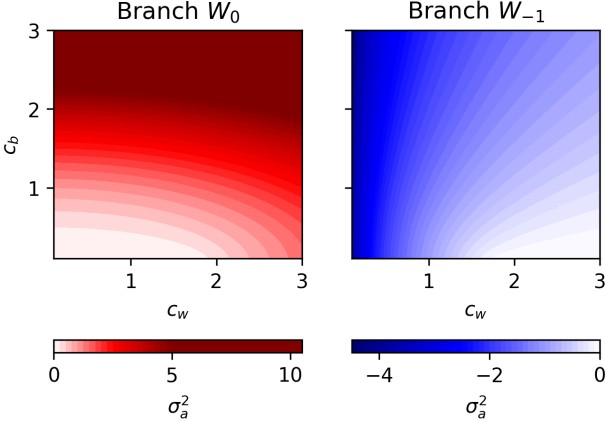

Figure 7: The $\sigma_a$ solution emerging from the $W_0$ branch on the left and $W_{-1}$ branch on the right

**Convergence Speed :** To quantify the convergence towards the fixed point $\sigma_a^2$, consider the derivative of $f$ at the fixed point:

$$f'(\sigma_a^2) = \frac{c_w^2}{3}e^{-2\sigma_a^2}.$$

The fixed point is exponentially attractive whenever $f'(\sigma_a^2) < 1$, which is immediately satisfied for $c_w < \sqrt{3}$. For $c_w > \sqrt{3}$, Lemma A.3 gives

$$f'(\sigma_a^2) = 2(-f(\sigma_a) + \frac{c_w^2}{6} + c_b^2) = -\mathcal{W}_0\left(-\frac{c_w^2}{3}e^{-c_w^2/3-2c_b^2}\right).$$

Since

$$-\frac{1}{e} < -\frac{c_w^2}{3}e^{-c_w^2/3-2c_b^2} < 0,$$

the properties of the principal branch $\mathcal{W}_0$ imply $|f'(\sigma_a^2)| < 1$. Hence, the fixed point is exponentially attractive for all values of $c_w \neq \sqrt{3}$, and convergence occurs rapidly. For $c_w = \sqrt{3}$, the map $f$ can be written

$$f(x) = \frac{1}{2}\left(1 - e^{-2x}\right), \qquad x \geq 0.$$

A Taylor expansion at $x = 0$ yields

$$f(x) = x - x^2 + \frac{2}{3}x^3 + O(x^4),$$

so that $f$ is tangent to the identity at the origin, i.e. $f(0) = 0$ and $f'(0) = 1$. Moreover, since $f(x) < x$ for all $x > 0$, the map $f$ admits $0$ as its unique fixed point on $[0, \infty)$, and any sequence $(\sigma_\ell)_{\ell \geq 0}$ defined by $\sigma_{\ell+1} = f(\sigma_\ell)$ with $\sigma_0 > 0$ is strictly decreasing and converges to $0$. Furthermore thanks to the previous extension it fits into the general class of one-dimensional parabolic maps studied in (Coll et al., 2020, Theorem 1). That theorem provides a complete asymptotic expansion of the orbit $(\sigma_\ell)$; in particular,

$$\sigma_\ell \sim \frac{1}{\ell} \qquad \text{as } \ell \to \infty.$$

This concludes the proof of the Lemma A.3, and of the Theorem 3.1. $\qquad\square$

$\square$

## A.2 GRADIENT DISTRIBUTION

**Theorem** (Restatement of Theorem 3.2). *Let $\mathbf{J}_\ell = \partial\mathbf{h}_\ell / \partial\mathbf{h}_{\ell-1}$ denote the Jacobian of the $\ell$-th layer. Under the same assumptions a Theorem 3.1 we have*

$$\mathbf{J}_\ell = \operatorname{diag}(\cos(\mathbf{z}_\ell))\,\mathbf{W}_\ell.$$

*In the limit of large $N$, each entry of $\mathbf{J}_\ell$ has zero mean and a sequence of variance $\widetilde{\sigma}_\ell^2$ such that the sequence $N(\widetilde{\sigma}_\ell^2)_{\ell \in \mathbb{N}}$ converges to*

$$\sigma_g^2 = \frac{c_w^2}{6}\left(1 + e^{-2\sigma_a^2}\right).$$

*Proof.* An element of the Jacobian of the $\ell$-th layer are written as:

$$\frac{\partial\mathbf{h}_{\ell,i}}{\partial\mathbf{h}_{\ell-1,k}} = \mathrm{W}_{\ell,i,k}\cos\left(\sum_{j=1}^{N}\mathrm{W}_{\ell,i,j}\mathbf{h}_{\ell-1,j} + \mathbf{b}_{\ell,i}\right) = \mathrm{W}_{\ell,i,k}\cos\left(\mathbf{z}_{\ell,i}\right)$$

with $\mathbf{z}_{\ell,i}$ the $i^{th}$ component of pre-activation vector defined in equation 3. In the limit of large width $N\infty$ $\mathbf{W}_\ell$ and $\mathbf{z}_\ell$ are independent (leave-one-out justification), resulting in the independence of variable $\mathrm{W}_{\ell,i,k}$ and $\cos(\mathbf{z}_{\ell,i})$. The variance of their product denoted $\widetilde{\sigma}_\ell^2$ can then be expressed as the product of their variance:

$$\widetilde{\sigma}_\ell^2 = \operatorname{Var}\left[\mathrm{W}_{\ell,i,k}\right]\operatorname{Var}\left[\cos\left(\mathbf{z}_{\ell,i}\right)\right].$$

Considering the same arguments as for Theorem 3.1 and replacing $\sin$ by $\cos$, the sequence $(N\widetilde{\sigma}_\ell^2)_{\ell \in \mathbb{N}}$ converges to

$$\sigma_g^2 = \frac{c_w^2}{6}(1 + e^{-2\sigma_a^2}),$$

with $\sigma_a^2$ the limit variance of the pre-activation, given by Theorem 3.1.

$\square$

## A.3 Proof of Equation 8 and Initialization 9

We propose to initialize the weights and biases of SIREN networks as follows:

$$\mathbf{W}_\ell \sim \begin{cases} \mathcal{U}\left(-\frac{\omega_0}{n_0}, \frac{\omega_0}{n_0}\right), & \ell = 1, \\ \mathcal{U}\left(-\frac{c_w}{\sqrt{N}}, \frac{c_w}{\sqrt{N}}\right), & \ell \in \{2, \dots, L\}, \end{cases}$$

and

$$\mathbf{b}_\ell \sim \mathcal{N}(0, c_b^2), \ \ell \in \{1, \dots, L\}.$$

To control the distribution scaling of gradients, following equation 11, we impose $\sigma_g^2 = 1$, i.e.,

$$\frac{c_w^2}{6}\left(1 + e^{-\sigma_a}\right) = 1. \tag{27}$$

Let's recall that the fix point $\sigma_a$ verifies :

$$\sigma_a^2 = \frac{c_w^2}{6}\left(1 - e^{-2\sigma_a^2}\right) + c_b^2$$

From equation 27 and , we easily get

$$c_b = \sqrt{1 - \frac{c_w^2}{3} - \frac{1}{2}\log\left(\frac{6}{c_w^2} - 1\right)}. \tag{28}$$

Combining this result with equation 27 leads to an implicit equation for $c_b^2$.

We discuss in the text two particular points, corresponding to $\sigma_a = 0$ and $\sigma_1 = 1$, respectively:

- The case $\sigma_a = 0$ (proposed initialization) leads to $(c_w, c_b) = (\sqrt{3}, 0)$.
- The case $\sigma_a = 1$ leads to $c_w^2 = 6/(1 + e^{-1})$. To obtain an explicit expression for $c_b$, it is convenient to use the fixed-point equation 26 with $x = 1$, leading to:

$$\frac{c_w^2}{6}\left(1 - e^{-2}\right) + c_b^2 = 1, \tag{29}$$

which, using equation equation 27, simplifies to

$$c_b^2 = \frac{c_w^2 e^{-2}}{3}. \tag{30}$$

## A.4 Derivation of the Proposed Scaling

Let $\Psi_\theta(\boldsymbol{x})$ defined as in equation 5 a scalar output function, initialized as in the previous theorems, and considering a given value of $\sigma_g$ resulting from the initialization.

**Derivation of the parameter-wise Gradient scaling:** Considering a weight-parameter $\mathbf{W}_{\ell,i,j}$ with $\ell > 1$ of the $\ell$-th layer, we study the scalar $\frac{\partial \Psi_\theta(\boldsymbol{x})}{\partial \mathbf{W}_{\ell,i,j}}$, which can be rewritten as :

$$\frac{\partial \Psi_\theta(\boldsymbol{x})}{\partial \mathbf{W}_{\ell,i,j}} = \frac{\partial \Psi_\theta}{\partial \mathbf{h}_{L-1}} \frac{\partial \mathbf{h}_{L-1}}{\partial \mathbf{h}_{L-2}} \cdots \frac{\partial \mathbf{h}_{\ell+1}}{\partial \mathbf{h}_\ell} \frac{\partial \mathbf{h}_\ell(\boldsymbol{x})}{\partial \mathbf{W}_{\ell,i,j}}$$

Then from theorem 3.2 under the choice of our initialization we know that the Jacobian matrices $\mathbf{J}_\ell = \partial \mathbf{h}_\ell / \partial \mathbf{h}_{\ell-1}$ have variance $\sigma_g^2/N$ in the limit of large $l$ and large $N$. Moreover, we have from the definition of $\Psi_\theta$ the expression of the vector $\frac{\partial \Psi_\theta}{\partial \mathbf{h}_{L-1}} = \mathbf{W}_L$ with $\mathrm{Var}(\mathbf{W}_L) \sim 1/N$. Let us consider first the sensitivity vector $\mathbf{g}_\ell$:

$$\mathbf{g}_\ell = \frac{\partial \Psi_\theta}{\partial \mathbf{h}_{L-1}} \frac{\partial \mathbf{h}_{L-1}}{\partial \mathbf{h}_{L-2}} \cdots \frac{\partial \mathbf{h}_{\ell+1}}{\partial \mathbf{h}_\ell}. \tag{31}$$

Owing to the impact of matrix multiplication on every components, we have $\text{Var}(\mathbf{g}_\ell) \sim (N\sigma_g^2)^{L-\ell-1}/N$. Let us now consider now the term $\frac{\partial \mathbf{h}_\ell(\boldsymbol{x})}{\partial \mathbf{W}_{\ell,i,j}}$. This is a zero vector except for the $i$-th component, verifying $\frac{\partial \mathbf{h}_{\ell,i}(\boldsymbol{x})}{\partial \mathbf{W}_{\ell,i,j}} = \mathbf{h}_{\ell-1,j}\cos(\mathbf{W}_{\ell-1,i,:}\mathbf{h}_{\ell-1} + \mathbf{b}_i)$, with variance $\text{Var}(\frac{\partial \mathbf{h}_{\ell,i}(\boldsymbol{x})}{\partial \mathbf{W}_{\ell,i,j}}) \sim 1$. Hence, the parameter-wise gradient can be rewritten as:

$$\frac{\partial \Psi_\theta(\boldsymbol{x})}{\partial \mathbf{W}_{\ell,i,j}} = \mathbf{g}_{\ell,i}\mathbf{h}_{\ell-1,j}\cos(\mathbf{W}_{\ell-1,i,:}\mathbf{h}_{\ell-1} + \mathbf{b}_i).$$

Assuming independence between $\mathbf{g}_{\ell,i}$ and $\frac{\partial \Psi_\theta(\boldsymbol{x})}{\partial \mathbf{W}_{\ell,i,j}}$, we finally obtain the desired variance scaling, namely $\text{Var}(\frac{\partial \Psi_\theta(\boldsymbol{x})}{\partial \mathbf{W}_{\ell,i,j}}) \sim (\sigma_g^2)^{L-\ell-1}/N$.

**Derivation of the input-wise Gradient scaling:** Following the same notations as above, we have:

$$\frac{\partial \Psi_\theta(\boldsymbol{x})}{\partial \boldsymbol{x}} = \frac{\partial \Psi_\theta(\boldsymbol{x})}{\partial \mathbf{h}_{L-1}}\frac{\partial \mathbf{h}_{L-1}}{\partial \mathbf{h}_{L-2}}\cdots\frac{\partial \mathbf{h}_2}{\partial \mathbf{h}_1}\frac{\partial \mathbf{h}_1(\boldsymbol{x})}{\partial \boldsymbol{x}}.$$

Recalling that $\mathbf{g}_1$, has variance $\text{Var}(\mathbf{g}_1) \sim (\sigma_g^2)^{L-2}/N$. In that case the $1/N$ factor will cancel out due to the term $\frac{\partial \mathbf{h}_1(\boldsymbol{x})}{\partial \boldsymbol{x}}$. Indeed, we have:

$$\frac{\partial \mathbf{h}_1(\boldsymbol{x})}{\partial \boldsymbol{x}} = \text{diag}(\cos(\mathbf{W}_1\boldsymbol{x} + \mathbf{b}))\,\mathbf{W}_1,$$

which is a non-trivial matrix of variance $\text{Var}(\frac{\partial \mathbf{h}_1(\boldsymbol{x})}{\partial \boldsymbol{x}}) \sim w_0^2$, for both the original and proposed SIREN initialization. Focusing on one input coordinate $x_i$, we get:

$$\frac{\partial \Psi_\theta(\boldsymbol{x})}{\partial x_i} = \mathbf{g}_1\,\text{diag}(\cos(\mathbf{W}_1\boldsymbol{x} + \mathbf{b}))\,\mathbf{W}_{1,:,i} = \sum_j \mathbf{g}_{1,j}\,(\text{diag}(\cos(\mathbf{W}\boldsymbol{x} + \mathbf{b}))\,\mathbf{W}_{1,:,i})_j.$$

The variance of each term scales as $\sim (\sigma_g^2)^{L-2}/N$. Supposing independence between each summand leads to $\text{Var}(\frac{\partial \Psi_\theta(\boldsymbol{x})}{\partial \boldsymbol{x}}) \sim (\sigma_g^2)^{L-2}w_0^2$.

# B  EXPERIMENTAL APPENDIX

## B.1  END TO END JACOBIAN, SINGULAR VALUE SPECTRUM

As discussed in (Pennington et al., 2017), an important notion of stability in neural networks is captured by the singular value distribution of the end-to-end Jacobian: when these singular values concentrate around 1, the network preserves the norm of signals during backpropagation. This property, known as *dynamical isometry*, is closely linked to stable and efficient training and will be the subject of further investigation for SIREN architectures in future work.

As a preliminary step toward this analysis, we plot figure 8 the full singular value distribution of the end-to-end Jacobian obtained with our proposed initialization. Since we focus on INR settings, we define the end-to-end Jacobian as the matrix of size $N \times N$, where $N$ denotes the width of the network:

$$\mathbf{J} = \frac{\partial \mathbf{h}_{L-1}}{\partial \mathbf{h}_1}$$

Once again, our initialization with $\sigma_a = 0$ exhibits a stable and nearly unitary normalized maximum singular value, independently of network depth. This behaviour is not observed for the other initialization schemes, where the largest singular value either grows steadily with depth or collapses rapidly, as in the case of the PyTorch initialization. However, our initialization does not achieve full dynamical isometry, indicating that there remains room for improvement while still satisfying the key constraints established earlier. Exploring additional constraints on the weight distribution may therefore lead to enhanced stability with respect to dynamical isometry.

## B.2  NTK SPECTRUM AND FOURIER OVERLAP

### B.2.1  NTK SPECTRUM

In the main text, we restricted our analysis of the Neural Tangent Kernel (NTK) spectrum to its trace, which captures only its mean behaviour. However, the trace alone does not reflect the full structure

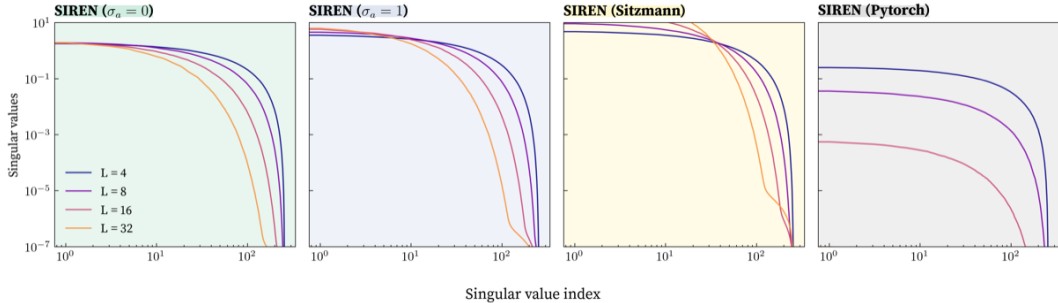

Figure 8: Full singular value spectrum evolution with depth for the proposed initializations $\sigma_a = 0$ and $\sigma_a = 1$, for the original Sitzmann initialization, and for the PyTorch default weight initialization. Each spectrum was averaged over five independently initialized networks. The Jacobian distribution was computed twice and averaged, using 10 sample points on the domain $[-\pi, \pi]$.

of the spectrum. In this section, we therefore examine the complete NTK eigenvalue distribution in order to highlight its finer characteristics.

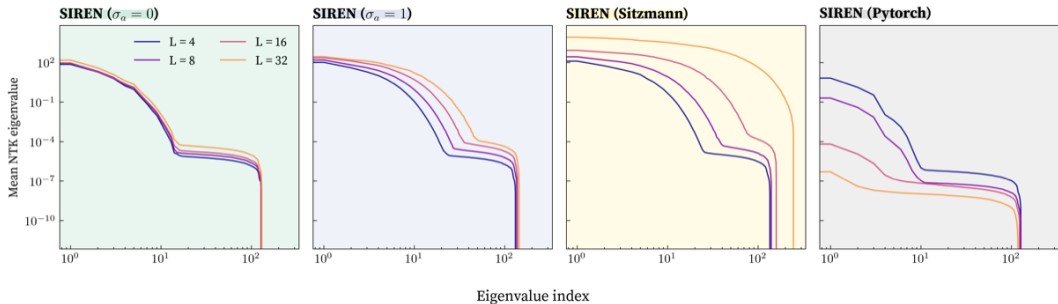

Figure 9: Full NTK eigenspectrum evolution with depth for the proposed initializations $\sigma_a = 0$ and $\sigma_a = 1$, for the original Sitzmann initialization, and for the PyTorch default weight initialization. Each spectrum was averaged over five independently initialized networks. The NTK was computed on the domain $[-\pi, \pi]$ using 256 sample points.

The full spectrum analysis shown figure 9 reinforces our previous observations based on the NTK trace, namely that the Sitzmann and PyTorch initializations become extremely ill-conditioned as depth increases. In contrast, the $\sigma_a = 1$ and $\sigma_a = 0$ initializations remain comparatively stable. One can observe a noticeable lifting of the eigenvalues at high indices for $\sigma_a = 1$, whereas this lifting is much smaller and more uniform under the $\sigma_a = 0$ initialization. This behaviour could be directly related to aliasing phenomena in such networks, where high frequencies can be used earlier to fit a signal.

This interpretation is further supported by the next analysis, where we show that under ill-conditioned initializations the low-index NTK eigenvectors begin to encode increasingly high frequencies as depth grows.

### B.2.2 Fourier Overlap

To support our NTK analysis and our explanation of spectral bias, we previously assumed (see Figure 5) a form of alignment between the eigenvectors of the SIREN NTK and the Fourier modes. To verify this assumption for our different initialization schemes, we examined the power spectrum of the NTK eigenvectors, which corresponds to their overlap with the Fourier modes:

$$|\langle \boldsymbol{v}_n, \phi_\omega \rangle|^2 = \left| \int_\Omega \boldsymbol{v}_n(x)\, e^{-i\omega x}\, dx \right|^2. \tag{32}$$

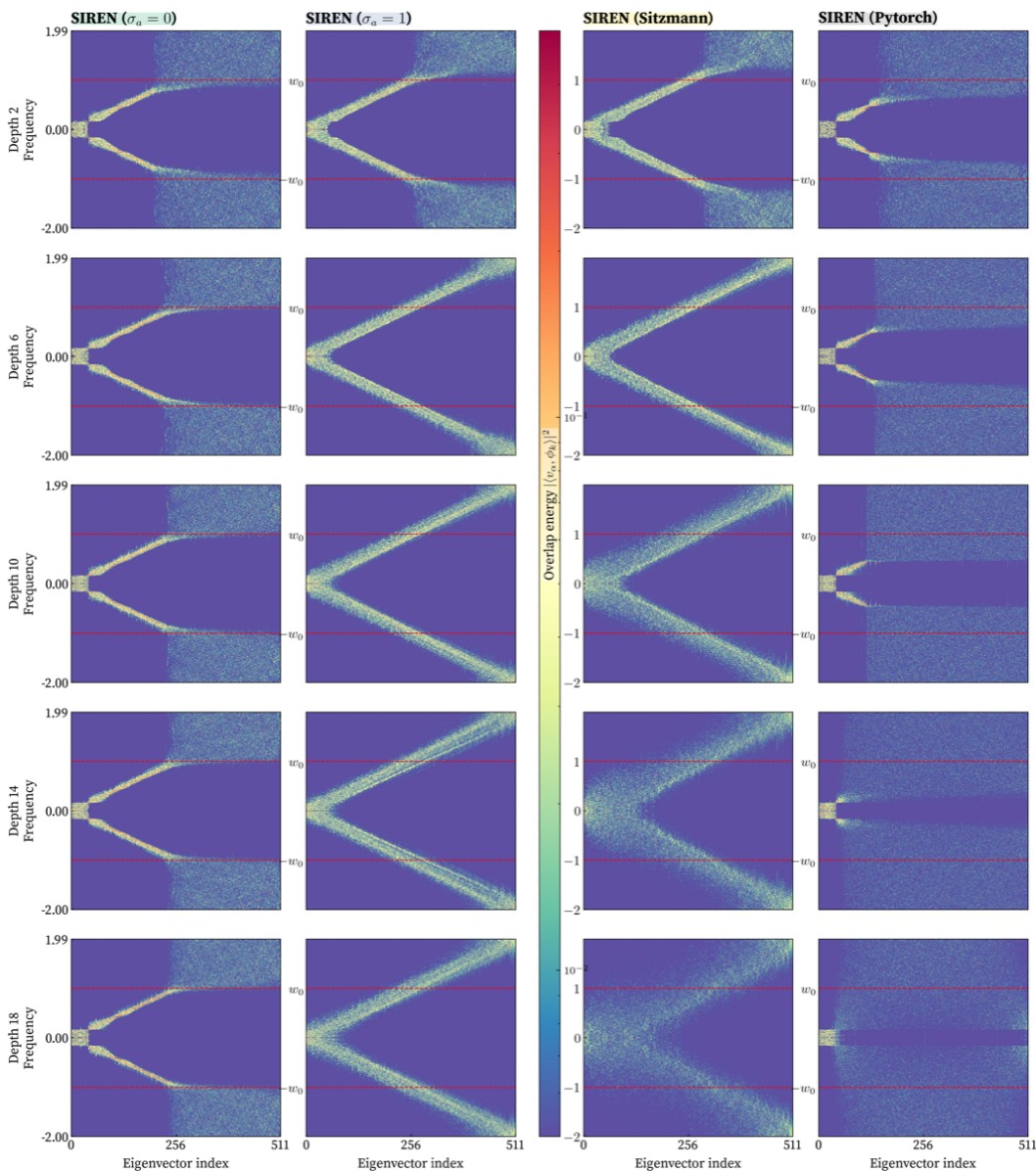

Figure 10: Overlap evolution with depth of the NTK eigenbasis over the Fourier modes, for the proposed initializations $\sigma_a = 0$ and $\sigma_a = 1$, the original Sitzmann initialization and the initialization with Pytorch default initialization weight. The power spectrum has been calculated using $w_0 = 1$, over the interval $[-64, 64]$ using 512 points. $w_0$ has been chosen to be two times smaller than the Nyquist frequency of the input points for the sake of vizualization. The horizontal red dashed lines correspond to the frequencies $\pm\omega_0$.

The previous analysis reveals that the only initialization preserving the expected ordering, *low frequencies* corresponding to *low NTK eigenvalues*, is our proposed initialization with $\sigma_a = 0$. This observation is consistent with our Fourier-spectrum study (see Section 3.3). Indeed, we observe in Figure 10 an almost perfect alignment between the Fourier modes and the NTK eigenspectrum for frequencies below $w_0$.

For the other initialization schemes, this alignment deteriorates substantially as depth increases, calling into question the relevance of NTK-based explanations of spectral bias. Indeed, in the NTK regime, the first modes learned are no longer the low-frequency components; instead, higher-frequency modes increasingly dominate for $\sigma_a = 1$ and the Sitzmann initialization. For the PyTorch

initialization, the situation is reversed: the entire spectrum collapses, preventing any meaningful frequency ordering.

## B.3 AUDIO FITTING EXPERIMENTS

To investigate the effect of the proposed initialization on the network's ability to fit high–frequency signals, we consider a 7-second audio clip sampled at the standard rate of 44,200 Hz. To expose potential generalization effects, we subsample the signal by a factor of three and set $w_0 = 7000$, which is approximately the Nyquist frequency corresponding to this reduced sampling rate. The results are shown figure 11.

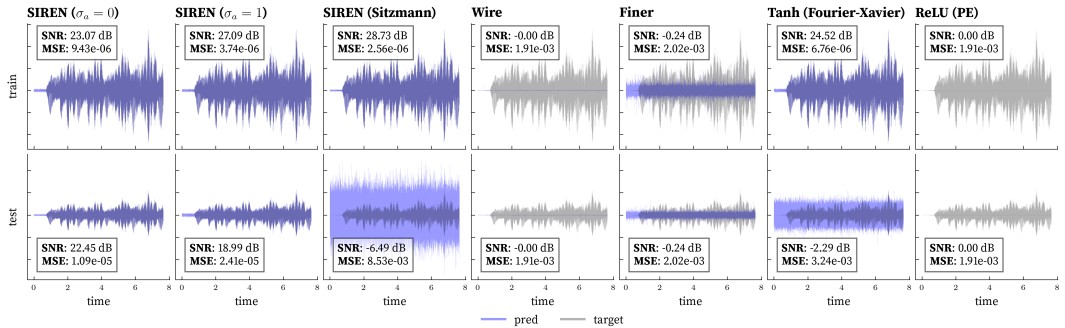

Figure 11: Comparison of several state-of-the-art methods (described in Figure 2) with SIREN using our proposed initialization. All networks, with depth $L = 15$ and width $N = 256$, were trained for 10,000 epochs using the ADAM optimizer with a learning rate of $3 \times 10^{-5}$.

Both the **SNR** and **MSE** metrics show a consistent improvement when using our proposed initialization on generalization tasks, while also providing strong training performance. The initialization with $\sigma_a = 1$ also achieves competitive results, though its generalization accuracy remains noticeably lower. For the other initialization schemes, even when training performance is satisfactory, the generalization error remains far too large to reliably encode a continuous signal.

## B.4 VIDEO FITTING EXPERIMENTS

**Video fitting on ERA-5 wind fields.** To evaluate the impact of the initialization on a complex video-fitting task, we consider the hourly ERA-5 atmospheric reanalysis on the spherical Earth, focusing on the 10 m meridional (South-North) wind component $v(t, \lambda, \varphi)$.

Where the data is defined on a regular longitude–latitude grid with

$$\lambda \in [0, 360), \quad \varphi \in [-90, 90],$$

discretized into

$$N_\lambda = 1440 \quad \text{and} \quad N_\varphi = 720$$

spatial points, respectively. We restrict ourselves to the first $T_{\max} = 30$ hourly time steps. For training, we form a set of input–output pairs

$$\left(\boldsymbol{x}_i, \boldsymbol{y}_i\right)_{i \in \mathbb{I}},$$

where each index $i$ corresponds to a triplet $(t, \lambda, \varphi)$ on this spatio-temporal grid. The target $\boldsymbol{y}_i$ is obtained from $v(t, \lambda, \varphi)$ by a standard affine normalization (subtracting a global mean and dividing by a global standard deviation computed over the first $T_{\max}$ frames).

Each input vector is defined as

$$\boldsymbol{x}_i = \left(\tau(t_i), \lambda_i, \varphi_i\right),$$

where the time coordinate $\tau(t)$ is obtained via a linear rescaling of the discrete time index $t$ such that the effective Nyquist frequency along the time axis matches that of the two spatial axes (longitude and latitude). This ensures a comparable frequency bandwidth in all three input directions and allows us to pick $w_0 = 0.7$ for every direction.

For training, we randomly subsample a fixed fraction of the full spatial gridded points $\{1, \ldots, N_\lambda\} \times \{1, \ldots, N_\varphi\}$ (10% of all points, justifying the choice of $w_0$), while for evaluation we use the complete spatio-temporal grid.

Regarding the batching, to avoid `I/O` bottlenecks when accessing the dataset, we organize the data into time-slice batches. Concretely, we consider a spatio-temporal grid

$$t \in \{0, \ldots, T_{\max} - 1\}, \qquad \lambda \in \{\lambda_1, \ldots, \lambda_{N_\lambda}\}, \qquad \varphi \in \{\varphi_1, \ldots, \varphi_{N_\varphi}\},$$

and for each fixed time index $t$ we form a batch containing many spatial points on the sphere. For a given time $t$, we define a (possibly subsampled) index set $\mathcal{I}_t \subset \{1, \ldots, N_\lambda\} \times \{1, \ldots, N_\varphi\}$, and construct the corresponding mini-batch

$$\mathcal{B}_t = \left\{ \left( \boldsymbol{x}_{t,j,k}, \boldsymbol{y}_{t,j,k} \right) : (j, k) \in \mathcal{I}_t \right\},$$

where each input is $\boldsymbol{x}_{t,j,k} = \left( \tau(t), \lambda_j, \varphi_k \right)$ and the target $\boldsymbol{y}_{t,j,k}$ is the normalized wind value at time $t$ and location $(\lambda_j, \varphi_k)$.

We benchmark previous state-of-the-art INR methods and our SIREN models with different initialization schemes on this ERA-5 re-analysis to assess their ability to fit and generalize complex spatio-temporal dynamics on the sphere.

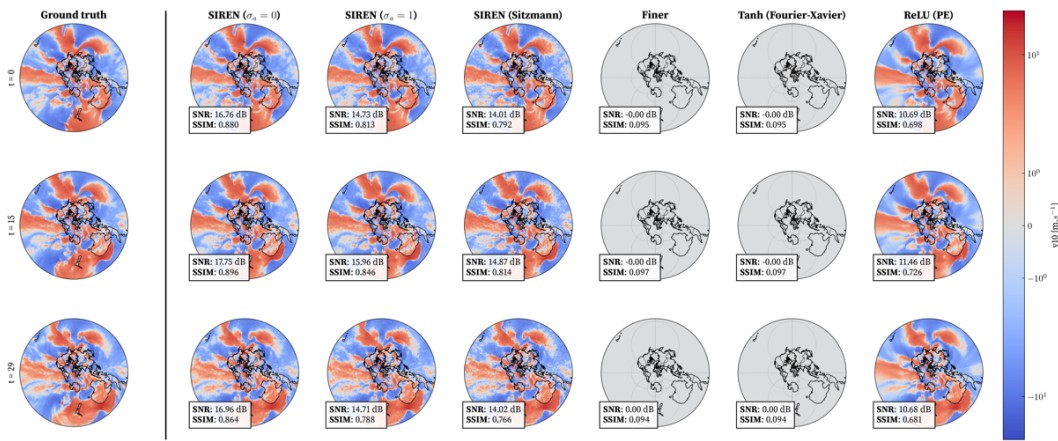

Figure 12: Comparison over three different time frames of several state-of-the-art methods on the ERA-5 reanalysis dataset (first 30 hours), using networks with width $N = 256$ and depth $L = 15$. All models were trained for 6,000 epochs with the `ADAM` optimizer and a *Reduce-on-Plateau* learning-rate scheduler, starting from an initial learning rate of $10^{-3}$. For batching, we used the time-slice structure described above with 5 gradient accumulation steps. To reduce computation time, we employed gradient scaling together with automatic mixed-precision (AMP) training.

.

Once again, our initialization with $\sigma_a = 0$ yields better generalization performance, even on complex tasks and geometries such as video fitting on the sphere. In contrast, the Sitzmann and $\sigma_a = 1$ initializations tend to produce noticeable noisy artifacts. Moreover, the `FINER` and `WIRE` methods appear clearly unstable for high-depth networks. We also highlight the comparatively good performance of the positional encoding `ReLU (PE)` network in this setting.

## B.5 DENOISING EXPERIMENTS

We consider a grayscale image $\boldsymbol{y}^\star : \Omega \subset \mathbb{R}^2 \to [0, 1]$ (the `astronaut` image), defined on a continuous domain $\Omega$. For training, we sample a regular grid of locations

$$(\boldsymbol{x}_i)_{i \in \mathbb{I}}, \qquad \mathbb{I} = \{1, \ldots, 128\} \times \{1, \ldots, 128\},$$

which we identify with points in $[-1, 1]^2$. The clean training targets are

$$\boldsymbol{y}_i = \boldsymbol{y}^\star(\boldsymbol{x}_i) \in [0, 1], \qquad i \in \mathbb{I}.$$

To study denoising and the implicit spectral regularization of different initializations, we corrupt only the training targets with synthetic high-frequency noise. Let $N = 128$ be the spatial resolution of the training grid and let

$$f_{\text{Nyq}} = \frac{N}{4}$$

denote the associated Nyquist frequency (in cycles per unit length on $[-1, 1]$). We construct a high-frequency noise field as a superposition of $K$ random waves whose spatial frequencies lie strictly above $f_{\text{Nyq}}$:

$$\eta(\boldsymbol{x}) = \sum_{k=1}^{K} \sin\Big(2\pi\big(f_x^{(k)} x_1 + f_y^{(k)} x_2\big) + \phi^{(k)}\Big),$$

where for each $k$ we draw $f_x^{(k)}, f_y^{(k)} \sim \mathcal{U}\big(2f_{\text{Nyq}}, 4f_{\text{Nyq}}\big)$, $\phi^{(k)} \sim \mathcal{U}(0, 2\pi)$, and $\boldsymbol{x} = (x_1, x_2)^\top$. We then normalize this field on the training grid to have zero mean and unit variance,

$$\tilde{\eta}_i = \frac{\eta(\boldsymbol{x}_i) - \frac{1}{|\mathbb{I}|} \sum_{j \in \mathbb{I}} \eta(\boldsymbol{x}_j)}{\sqrt{\frac{1}{|\mathbb{I}|} \sum_{j \in \mathbb{I}} \big(\eta(\boldsymbol{x}_j) - \frac{1}{|\mathbb{I}|} \sum_{\ell \in \mathbb{I}} \eta(\boldsymbol{x}_\ell)\big)^2}}, \qquad i \in \mathbb{I},$$

and scale it by a prescribed noise level $\sigma_{\text{noise}} > 0$. The noisy training targets are finally defined as

$$\tilde{\boldsymbol{y}}_i = \boldsymbol{y}_i + \sigma_{\text{noise}} \tilde{\eta}_i, \qquad i \in \mathbb{I},$$

We train all INR models on the noisy dataset $\{(\boldsymbol{x}_i, \tilde{\boldsymbol{y}}_i)\}_{i \in \mathbb{I}}$ and evaluate on a higher-resolution grid covering the full image domain, using the clean image $\boldsymbol{y}^\star$ as reference. This setup isolates the ability of each initialization to act as an implicit frequency-space regularizer for denoising, independently of network depth.

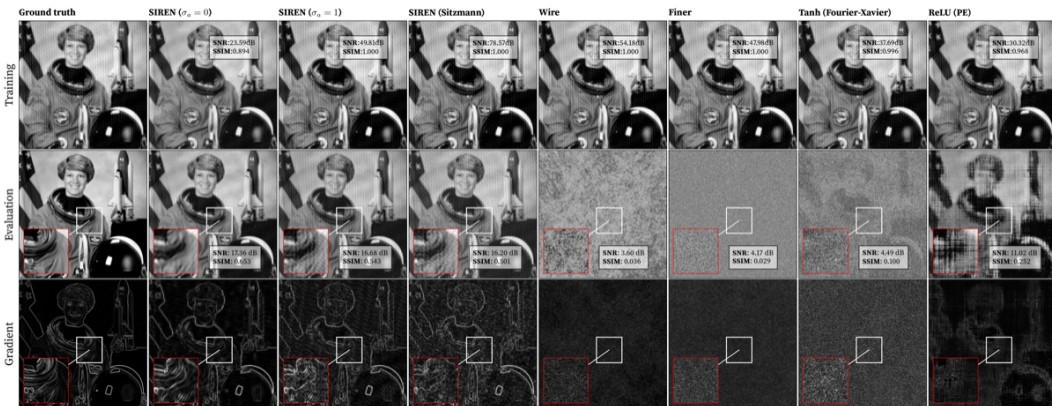

Figure 13: Results of the denoising experiments for the different state-of-the-art methods, using networks with width $N = 256$ and depth $L = 10$. All models are trained on the noisy dataset $\{(\boldsymbol{x}_i, \tilde{\boldsymbol{y}}_i)\}_{i \in \mathbb{I}}$ described above using $\sigma_{\text{noise}} = 0.05$ and evaluated on the original high-resolution image of size $512 \times 512$ to assess denoising performance. The networks were trained for 10 000 epochs using the ADAM optimizer with a learning rate of $10^{-4}$.

Figure 13 illustrates our claim that the proposed initialization acts as a regularizer on the frequency content that the network can represent. Indeed, we observe higher **SNR** and lower **MSE** for our initialization $\sigma_a = 0$, together with a significantly larger training loss. This indicates that the network does not fit all of the high-frequency background noise, but instead focuses on reconstructing the underlying clean signal.

## B.6 PHYSICS INFORMED EXPERIMENTS

Physics-Informed Neural Networks (PINNs) approximate the solution $\boldsymbol{u}$ of a differential equation with $\Psi_\theta$ by embedding the underlying physical laws into the loss function. Given a PDE of the form

$$\mathcal{N}[\boldsymbol{u}](\boldsymbol{x}) = f(\boldsymbol{x}), \qquad \boldsymbol{x} \in \Omega,$$

with boundary/initial conditions $\mathcal{B}[u] = g(x)$ on $\partial\Omega$, the neural network $\Psi_\theta$ is trained by minimizing the composite loss

$$\mathcal{L}(\theta) = \lambda_f \sum_{x_f \in \mathcal{D}_f} |\mathcal{N}[\Psi_\theta](x_f) - f(x_f)|^2 \; + \; \lambda_b \sum_{x_b \in \mathcal{D}_b} |\mathcal{B}[\Psi_\theta](x_b) - g(x_b)|^2 .$$

where $\mathcal{D}_f$ and $\mathcal{D}_b$ denote collocation points in the domain and on the boundary. Automatic differentiation is used to compute $\mathcal{N}[\Psi_\theta]$, allowing the network to satisfy the governing equations as part of the training process.

In order to compare the several model at stake and the impact of the initialization, we used the PINNacle benchmark (Hao et al., 2024), which allowed us to have a pre-builtin solver for each differential equation we studied.

### B.6.1 BURGER 1D

We consider the one-dimensional viscous Burgers equation, written in the generic PDE form

$$\mathcal{N}[u](x,t) = u_t + u\,u_x - \nu u_{xx} = 0, \qquad (x,t) \in \Omega \quad, \nu = \frac{0.01}{\pi}.$$

The spatio-temporal domain is defined as $\Omega = [-1, 1] \times [0, 1]..$ The initial and boundary conditions are given by $u(x, 0) = -\sin(\pi x), \qquad u(-1, t) = u(1, t) = 0.$

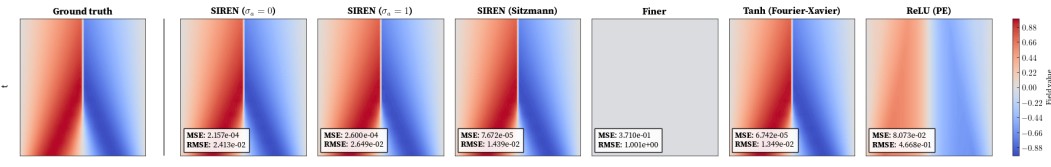

Figure 14: Results of the Burgers 1D solutions for the different state of the art methods, using a network with width $N = 256$ and $L = 15$. The networks were trained for 10 000 epochs using the ADAM optimizer with a learning rate of $10^{-4}$. For the SIREN based architectures, we chose $w_0 = 2$.

We observe figure 14 that the different initialization schemes yield very similar results, with the exception of the FINER and ReLU networks. Interestingly, for this specific task, the original Sitzmann initialization appears to provide the most favorable performance. We conjecture that this behavior is related to the nature of the Burgers equation, whose sharp propagating front can be effectively represented even under a highly ill-conditioned gradient distribution.

### B.6.2 STATIONARY NAVIER-STOKES 2D

We consider the stationary incompressible 2D Navier-Stokes equations

$$\mathcal{N}_u[u, p] = (u \cdot \nabla)u + \nabla p - \nu \Delta u = 0, \qquad \mathcal{N}_p[u] = \nabla \cdot u = 0,$$

for the velocity field $u = (u, v)$ and pressure $p$, with $\nu = 1$.

The spatial domain $\Omega$ is defined as

$$\Omega = ([0, 8]^2) \setminus \bigcup_i R_i,$$

where each $R_i$ denotes a circular obstacle. For further details about the boundary conditions please see the original PINNacle benchmark (Hao et al., 2024).

The impact of initialization observed figure 15 is far more pronounced in that case than for Burger. We observe that having proper control over the spectral properties of the initialization can lead to a significant improvement in performance. The Sitzmann initialization exhibits, as expected, problematic high-frequency components, while other models such as FINER, Tanh, and ReLU fail completely to reconstruct the physical solution.

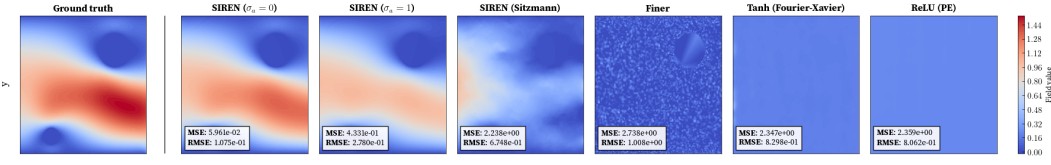

Figure 15: Results of the Navier-Stokes 2D solutions for the different state of the art methods, using a network with width $N = 256$ and $L = 15$. The networks were trained for 10 000 epochs using the ADAM optimizer with a learning rate of $10^{-4}$. For the SIREN based architectures, we chose $w_0 = 2$.

### B.6.3 HEAT EQUATION IN COMPLEX GEOMETRY

We consider the transient 2D heat equation

$$\mathcal{N}[u](\boldsymbol{x}, t) = u_t - \Delta u = 0, \qquad (\boldsymbol{x}, t) \in \Omega \times [0, 3].$$

The spatial domain $\Omega$ is defined as

$$\Omega = ([-8, 8] \times [-12, 12]) \setminus \bigcup_i R_i,$$

where each $R_i$ denotes a circular obstacle. For further detail about the boundary conditions please see the original PINNacle benchmark (Hao et al., 2024).

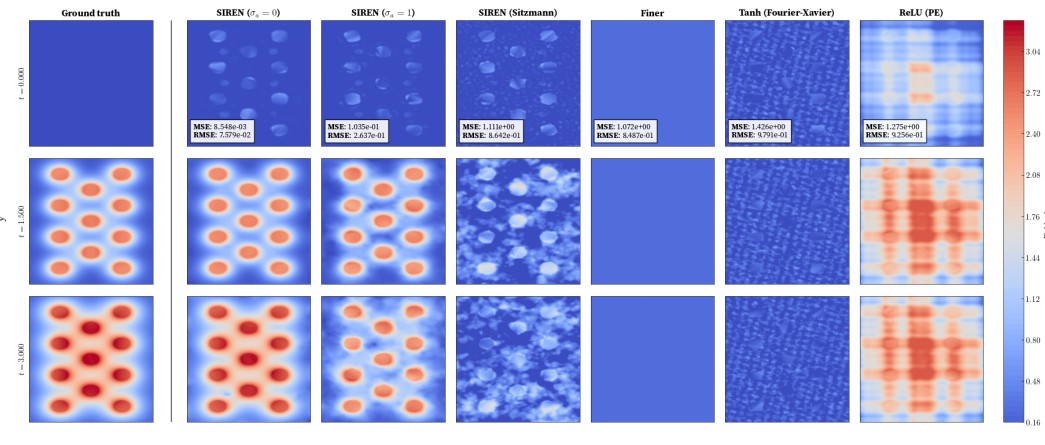

Figure 16: Results of the 2D heat equation experiments for the different state of the art methods, using a network with width $N = 256$ and $L = 15$. The networks were trained for 10 000 epochs using the ADAM optimizer with a learning rate of $10^{-4}$. For the SIREN based architectures, we chose $w_0 = 1$.

The results for different initializations are shown figure 16. The distinction between $\sigma_a = 1$ and $\sigma_a = 0$ is striking. The former produces noticeably noisy and unstable solutions, whereas setting $\sigma_a = 0$ successfully reproduces the behavior of the ground-truth solution. For the other initialization methods, the observations are consistent with those made in the Navier–Stokes experiment.

### B.7 SYNTHETIC EXPERIMENTS

### B.7.1 1D FITTING EXPERIMENTS

For the 1D fitting experiments, we generated synthetic data by sampling from a multi-scale function:

$$f_{1d}(x) = \sin(3x) + 0.7\cos(8x)$$
$$+ 0.3\sin(40x + 1) + \exp(-x^2)$$

To explore the impact of initialization on the performance of various neural network architectures, we studied two tasks: function fitting and PDE solving. Since image and video fitting reduce to function fitting, we focus on it. This choice lets us control the target function's frequency content. As a result, we can probe the different scales present in the data.

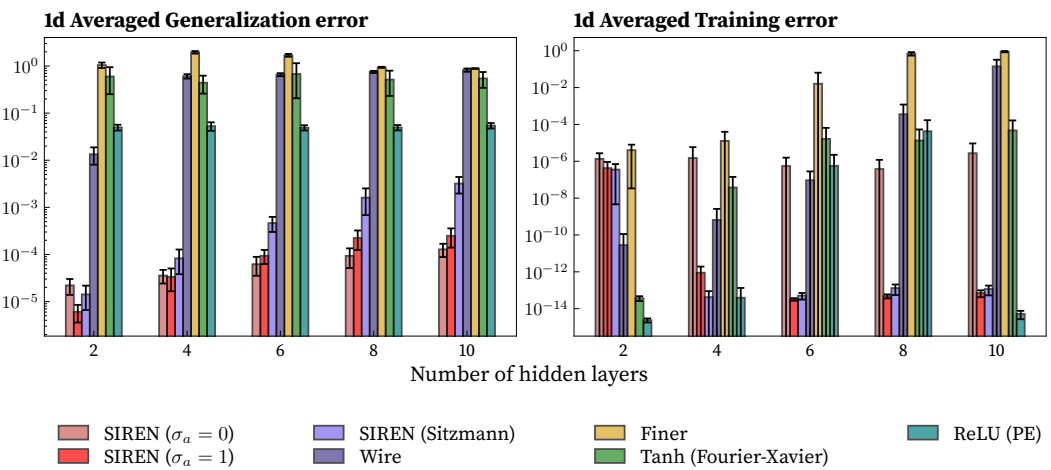

Figure 17: 1d Averaged generalization and training error for the 1D fitting problem. The results are averaged over 10 runs for each architecture of width $N = 128$. The error bars represent the standard deviation of the results.

The results plotted figure 17 show that our proposed initialization matches or exceeds the accuracy of the traditional SIREN architecture for fitting a function. Moreover, it delivers significantly lower generalization error compared to the original SIREN. Notably, the Tanh-based positional-encoding network also shows strong generalization performance, despite its slightly higher training error.

### B.7.2 2D FITTING EXPERIMENTS

We applied the same methodology to a two-dimensional, multi-scale test function:

$$f_{2d}(x, y) = \sin(3x)\cos(3y) + \sin(15x - 2)\cos(15y)$$
$$+ \exp\big(-(x^2 + y^2)\big),$$

for $(x, y) \in [-1, 1]^2$. The exponential term ensures no architecture can represent the function trivially. We sampled 3600 random training points, giving a Nyquist frequency above 15. Each network was trained for 5000 epochs using Adam (learning rate $10^{-4}$) under various initialization schemes. We then evaluated generalization error on 10 000 test points. The comparative results appear in Fig. 18.

The results mirror the 1D fitting experiments. Our proposed initialization clearly outperforms all other architectures on the generalization task. At the same time, it maintains a very low training error, comparable to the SIREN architecture.

### B.7.3 3D FITTING EXPERIMENTS

For the 3D fitting experiments, we use the same framework as in 1D and 2D. We test a three-dimensional function with multi-scale features:

$$f_{3d}(x, y, z) = \sin(5x)\cos(12y)\sin(3z)$$
$$+ \exp\big(-(x^2 + y^2 + z^2)\big),$$

for $(x, y, z) \in [-1, 1]^3$. The exponential term prevents trivial representation by any architecture. We sample 8000 random training points, ensuring a Nyquist frequency above 12. Each network trains for 5000 epochs using Adam with learning rate $10^{-4}$ under various initialization schemes. We then evaluate generalization error on 70 000 test points. The results appear in Fig. 19.

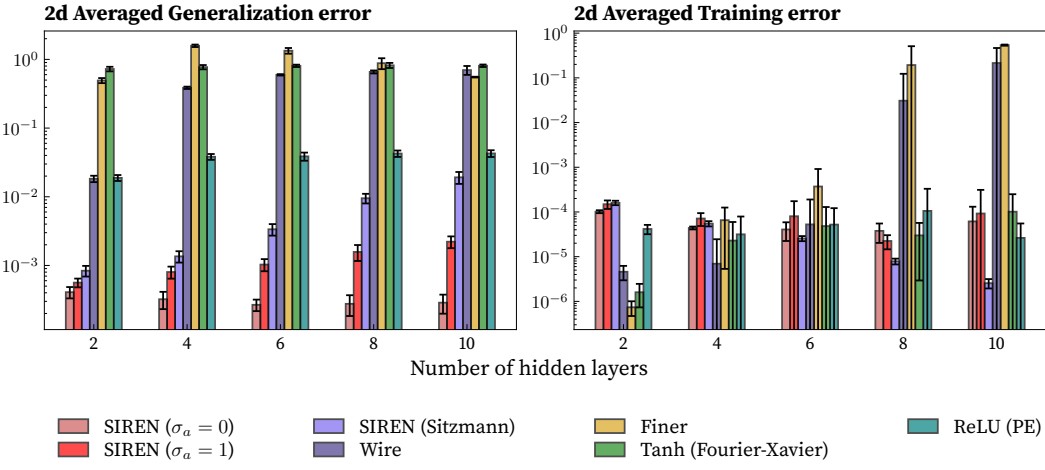

Figure 18: 2d Averaged generalization and training error for the 2D fitting problem. The results are averaged over 10 runs for each architecture of width $N = 1238$. The error bars represent the standard deviation of the results.

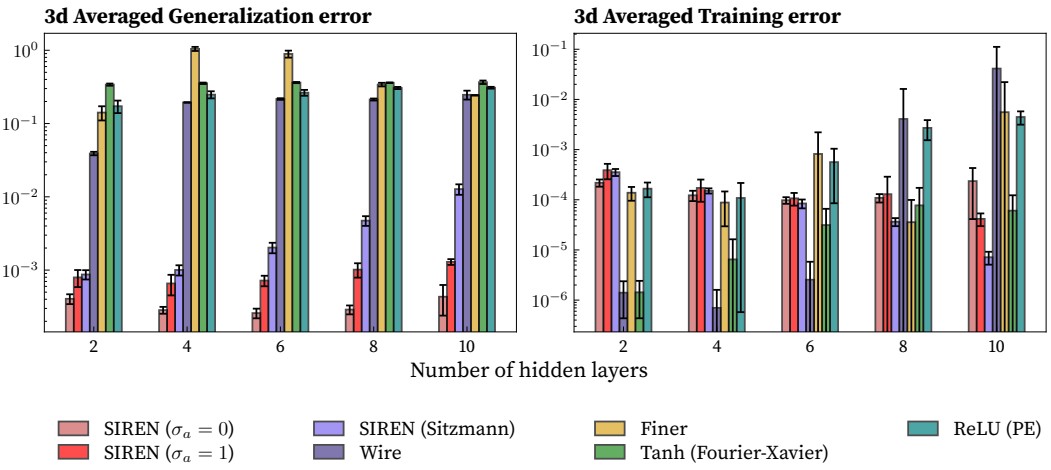

Figure 19: 3d Averaged generalization and training error for the 2D fitting problem. The results are averaged over 10 runs for each architecture of width $N = 128$. The error bars represent the standard deviation of the results.

Once again, our proposed initialization delivers strong results. It clearly outperforms all other architectures on generalization. Its fitting error remains very low, only slightly above the classic SIREN. Interestingly, as the number of layers increases, SIREN's training error decreases alongside rising high-frequency content. This suggests that fitting high frequencies may harm generalization—a drawback our method avoids.

## C  ABLATION STUDIES

Since our theoretical analysis is derived in the infinite-width and infinite-depth regime, we also evaluate our model in the opposite setting: using small widths and very large depths. This allows us to examine, on one hand, how finite-size effects modify the experimental behaviour, and on the other hand, whether our theoretical predictions remain valid when the depth becomes extremely large. This analysis further reveals how these factors influence the overall performance of such neural networks.

## C.1 FINITE WIDTH EFFECT

The finite-width experiment (with $N = 32$) leads to the same conclusions as the theoretical study: deep networks initialized with the Sitzmann scheme or with $\sigma_a = 1$ exhibit a high noise level. In contrast, our proposed initialization maintains a lower noise level (see the gradient section of Figure 20), even for very small widths, although it severely harms performance.

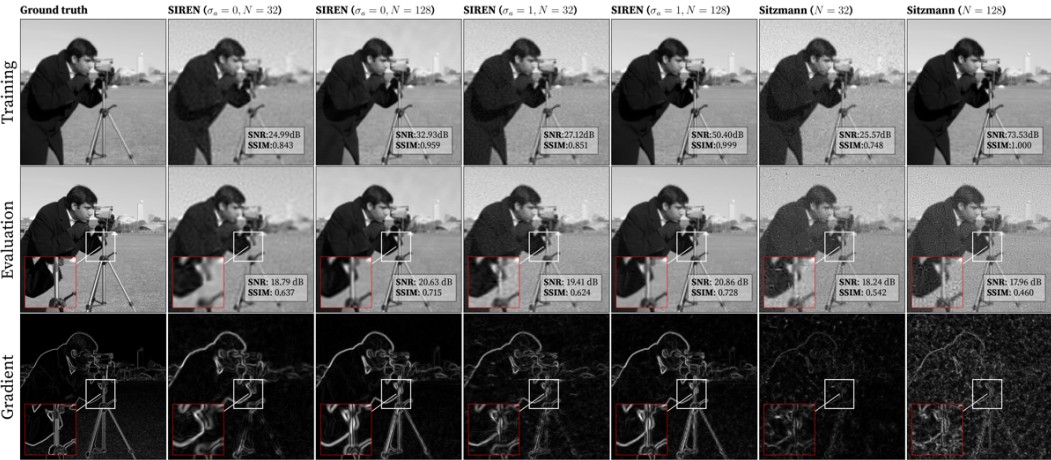

Figure 20: Comparison of the discussed initialization method, and how finite width ($N = 32$ and $N = 128$) affects their performance and behavior. The setting of the experiments are the same as one described in Figure 2

## C.2 LARGE DEPTH EFFECT

As shown Figure 21, the large-depth experiments with $L = 10$ and $L = 40$ confirm our previous theoretical discussion in the infinite-depth limit. In the case $\sigma_a = 0$, increasing the depth to $L = 40$ even improves performance and further reduces the effective noise level. For $\sigma_a = 1$, the performance at large depth is surprisingly good, despite the clear growth of high-frequency components with depth observed in the Fourier spectrum (see Figure 4); this observation still holds at $L = 10$. We attribute this behaviour to the long training time. For the Sitzmann original initialization, as expected, the increasing of depth severely impacts the generalization performances, due to overwhelming presence of high frequency components.

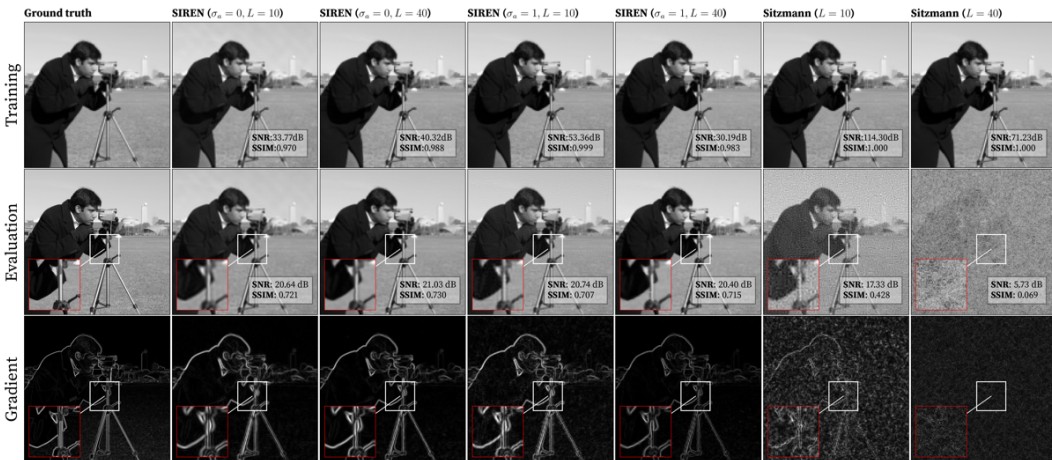

Figure 21: Comparison of the discussed initialization method, and how large depth affect their performance and behavior. The setting of the experiments are the same as one described in Figure 2

