# OpenReview forum: "A New Initialization to Control Gradients in Sinusoidal Neural Networks"
_ICLR.cc/2026/Conference — ICLR 2026 Poster_

### Official Review · Reviewer_Le3d · 2025-10-27

**Soundness:** 4
**Presentation:** 3
**Contribution:** 2
**Rating:** 4
**Confidence:** 4

**Summary:**

This paper studies the problem of vanishing/exploding gradient of sinusoidal neural networks at initialization. Based on theoretical analysis, it argues that the prior standard initialization scheme should be adjusted by an additional factor to prevent the vanishing/exploding gradient issue. Its empirical results on an image-fitting problem and NTK analysis support this new initialization strategy.

**Strengths:**

1: The new initialization method is theoretically justified to be better in stabilizing the gradient when network depth is large.

2: The efficacy of the new method is empirically verified in a real-world problem, and is also supported by the numerical analysis of the NTK matrix.

3: The new initialization method is simple to implement and requires small extra computation.

4: The paper is well-presented and is easy to follow.

**Weaknesses:**

1: The paper claims a causal relation between the gradient control and (spurious) high-frequency modes. For example, in abstract (Line 21) and introduction (Line 50, 55). However, I don’t quite see a proof or analysis of this causal relation in the paper. It seems the most related is Eq. 12, but this equation only shows a correlation, not causality. In addition, this correlation is loose: a stable gradient not necessarily tied with no spurious high frequency, as it also depends on $\omega_0$.

2: The scope of the paper is confined to the setting of $sin$ activation function. Does it extend to commonly used activation functions, such as ReLU? I know there are some empirical results on the image-fitting problem in Figure 2 for other activations. I’d like to know whether it extends beyond this problem, such as image classification, image generation where ReLU (or similar) are much more often used than $sin$. In addition, I’d like to see whether the theory extends.

3: It would be better if there is more empirical evidence, in addition to the image-fitting problem.

4: The paper exceeds the page limit by approximately one-third of a page and includes an acknowledgment section that may partially reveal the authors’ identities.

If the above are addressed, I am happy to increase the rating

**Questions:**

See comments in weaknesses

---

> ### Author Response · Authors · 2025-11-24
> **First Reply**
>
> ### **Weaknesses**
>
> - *1: The paper claims a causal relation [...] but I don’t see a proof; correlation seems loose and tied to $w_0$.*
>
>   - **Response:** This remark was particularly valuable, as it made us realize that our initialization does not by itself guarantee proper control of the frequency spectrum: controlling the gradients is not equivalent to controlling the spectrum, despite a strong correlation. We therefore added a dedicated section (p. 7–8) on this point and modified the initialization scheme. Concerning the dependence on $w_0$, we made this dependence explicit in that section; however, $w_0$ is a network hyperparameter that simply rescales the first-layer weights, is kept fixed in our experiments, and should in practice be chosen to match the specific problem at hand. We advise to take it as the nyquist frequency of the sampling we are using.
>
> - *2: The scope is limited to $\sin$ activations [...] does it extend to ReLU or other domains? Does the theory extend?*
>
>   - **Response:** Our theoretical results apply specifically to the sine activation function, which is widely used in INR tasks and covers a broad range of applications (for which we provide an extensive set of experiments in the experimental appendix, p. 23–29). For any domain where sine activations are employed, which is to our knowledge only used in INR setting, our analysis guarantees normalized gradient backpropagation as depth increases. If one is interested in other activation functions—such as ReLU, sigmoid, or tanh—there already exists substantial literature on gradient-normalized initialization at the Edge of Chaos (EoC), notably the work *Mean Field Residual Networks: On the Edge of Chaos* by Greg Yang and Samuel S.~Schoenholz.
>
> - *3: More empirical evidence beyond the image-fitting problem would help.*
>
>   - **Response:** We added an extensive list of experiments from audio (p. 23), video fitting (p. 24) to denoising (p. 25). We would also precise that some synthetic (1d, 2d and 3d) experiments were included in Appendix B (p. 26–27) to justify the benchmark (figure 1). We also plan to add some Physics Informed tasks to demonstrate stronger stability in the upcoming days.
>
> - *4: The paper exceeds the page limit and includes an acknowledgment revealing identities.*
>
>   - **Response:** In the author guidelines, it was specified that the reproducibility statement does not count toward the page limit. We also removed the acknowledgements section prior to the reviewer decision.

---

> > ### Comment · Reviewer_Le3d · 2025-11-27
> >
> > I thank the authors for the further clarification and revisions made.
> >
> > It seems that the proposed initialization method significantly changed during this rebuttal period. The newly proposed method requires the hidden layer features to converge to a fix point 0 as depth grows (rather than staying with variance 1, as proposed in the original submission). Two concerns: 1, this implies that the final layer's feature is ultimately indistinguishable regardless the input, which may lead to a trivial (or at least a decreased) prediction performance; 2, in this new setting, the network is probably out of the NTK regime, hence, it is questionable that whether the NTK theory can sill be applied for the theoretical analysis of the paper, such as in Section 4.
> >
> > My original concern regarding the (causal) relation between the gradient control and (spurious) high-frequency modes is still unclear, which should be a key part of the paper.
> >
> > Regarding my second point (sin activation vs general activation): As similar results are already known for other activations, what is new in this paper for sin activation, how significantly it differs from those.

---

> ### Author Response · Authors · 2025-11-27
> **Second Reply**
>
> - The core theoretical result of the paper is the expression for the variance of the pre-activations $\sigma_a\$ and the Jacobian entries $\sigma_g $ in the limit $N, L \to \infty $, expressed as a function of the network’s bias and weight initialisation (i.e., $c_b, c_w$). We emphasise that these results have not changed between the original and the revised versions.
>
> - Initially, we focused our analysis on the Sitzmann prescription ($\sigma_a = 1$) and mentioned in the conclusion the possibility of studying other choices of $\sigma_a$. Following several reviewer remarks requesting an investigation of the causal relation between gradients and frequencies, we extended our analysis to the frequency spectrum of the neural network. This led us to keep, as before, $\sigma_g = 1$, while using $\sigma_a = 0$ instead of $\sigma_a = 1$. This constraint, together with the previously derived results (Sections 3.1 and 3.2), led to the proposed initialisation.
>
> - We then show experimentally that this new choice retains the advantages of the previous one (which still outperforms other initializations strategy) and also provides better control of the frequency spectrum, revealing deeper stability while depth increase (section 3.3, p.7).
>
> - In the second updated version (introduction and abstract), we removed **every statement** suggesting a causal link between high gradients and high frequencies. We also clarify in the main text that controlling gradients and controlling frequencies are not equivalent (see Fig. 4). We also highlighted that the theoretical analysis of the spectrum requires deeper investigation (section 3.3, p.7) .
>
> - Regarding performance, we believe that our extensive set of experiments provides strong empirical support for initialisation choices based on Theorems 3.1–3.2 ($\sigma_a = 0, 1$) (see Appendix B.3, B.4, B.5, B.7, and the ablation study in C.1).
>
> - To our knowledge, for activation functions in MLPs other than ReLU (whose case is trivial), there is no closed-form formula for the Edge-of-Chaos initialization (*On the Impact of the Activation Function on Deep Neural Networks Training*). Determining this boundary typically requires numerical computation. In this paper, we derive the analytic expression of this boundary, which is well defined for $\sin$-based networks (Proposition 3.1). We did not highlight this as a contribution because we do not consider it to be an important practical point.
>
> - Furthermore, when compared with a classical ReLU Kaiming at the EoC, in the first version of the article, we obtained much more interesting results.
>
> - In addition to that, the case $\sigma_a = 0$ has already been studied in several contexts, notably for the sigmoid activation (see *Resurrecting the Sigmoid in Deep Learning through Dynamical Isometry: Theory and Practice*).
>
> - Regarding the NTK, this is not part of the core theoretical results and consists only of a simple scaling analysis. Furthermore, changing $\sigma_a$ does not affect the validity of this theory and even strengthen the choice of $\sigma_a = 0$ (see Appendix B.2.1 & B.2.2).

---

### Official Review · Reviewer_1Ux1 · 2025-10-30

**Soundness:** 2
**Presentation:** 3
**Contribution:** 3
**Rating:** 4
**Confidence:** 2

**Summary:**

The paper proposes a new initialization for INRs that rely on sinusoidal activation functions. Choosing the parameters of this new initialization is theoretically motivated by avoiding vanishing/exploding gradient variances of the network with respect to the network depth and width. This is done by a series of theoretical results relying on NTK and assuming it is constant at the beginning of training. The result is the avoidance of spurious frequencies and other quality degradation that occurred otherwise when increasing the depth of the network using previous types of initialization. Empirical evaluations show better generalization when using the proposed initialization.

**Strengths:**

The paper’s strengths are:
- The paper studies a relevant problem which is theoretically motivated and has practical applications.
- The paper’s motivation and contributions are overall clear.
- The paper makes good connections with existing literature in both its applied and theoretical derivations.

**Weaknesses:**

I have a series of things to point out about the paper, which are the reason for the current score assigned to the paper.

**>>About the results:**
- It is my understanding that the SIREN initialization does not work well for large depths, and it is argued that the initialization by the authors (equation (7)) does not suffer from it. Well, this is really hard to know *for sure*: I understand that the theoretical results were derived in the large depth limit and using the NTK at initialization, but in practice, we are not in the NTK regime across training, so things could be very different. It would be important to try experiments where one keeps increasing the network’s depth using the authors’ initialization and see if there is a **tipping point** after which larger networks lead to deterioration on generalization tasks (due to spurious high frequencies or another cause). This is largely unexplored in the paper. If such tipping point is found, it will establish an important limitation on the practicality of the authors’ method. One could compare such a tipping point to the value of depth over which SIREN leads to deterioration—would such tipping point be x2, x10, etc. magnitude larger?
- There is something which is not clear at all to me: the meaning of the expression “limit of large $N$” in Theorem 3.2. When talking about limits, either you (i) have $N\to C$ where $C$ is a constant that is much larger than some other parameters, or you (ii) have $N\to\infty$. I believe Theorem 3 refers to item (i), since item (ii) does not make much sense because the equation of $\sigma_g^2$ depends on $N$. Then, what are the other parameters that are less than the constant $C$ that I defined above? I hope my question makes sense. The expression “limit of large $N$” **makes no sense** unless **you specify what makes $N$ “large”**. This is important to know for the paper.
- As a follow-up to the previous point: is equation (12) also under the “limit of large $N$”? If so, it would be good to make it explicit in the text and respond to my previous point — otherwise, a reader **can’t know what makes $N$ large enough for these expression to hold**.
- Another problem is that Theorem 3.2’s  results are under the same assumptions as Theorem 3.1.—i.e., assuming the **random initialization** proposed by the authors. Thus, the results of Theorem 3.2 such as the expression of $\sigma_g^2$ **must hold under some probability**. Can the authors clarify the probability under which these results hold? This is also very important for the formality of the results—until now, the reader can’t know if the results hold with probability $1$ or some probability dependent on problem parameters (such as the network width $n_0$, for example).

**>>Motivation:**
- In the last paragraph of the contributions in Section 1.3, it is mentioned that various results from the paper “will be valuable beyond the specific case of INR with sinusoidal activations”. Can the authors list examples of other cases beyond “INR with sinusoidal activations” for which their results are valuable and informative? All the results derived in the paper are highly specific to the setting of INR with sinusoidal activations, how is it then that the paper’s results can be transferred to other application domains?
- Lines 175-178: why is it important to ensure differentiability in the applications mentioned therein?
- The paper focuses on INR and SIREN, nonetheless, there is not much mention about areas (with citations) where it has been successfully applied. I see that PDE-related applications are mentioned (without providing a citation), but I am sure there are more applications of interest. This will strengthen the motivation of the paper.
- In Section 4, a considerable assumption is made: that the NTK remains constant during training. Nonetheless, this, to my best understanding, is not something that will commonly hold in practice. The paper seems to claim this is the case, since it mentions “the early training dynamics is fully determined by the spectral properties of the NTK at initialization”. Can the authors provide references/citations or a more formal justification for such a claim? If I were to make a guess, networks with very large width are the ones that more likely will preserve the spectral properties of the NTK given at initialization after a few iterations of gradient descent with a small learning rate (but this is a guess, the authors should look at the literature or provide a good formal justification).
- In the last paragraph of Section 5 it is mentioned that the “pre-activation variance ($\sigma_a^2=1$) may be adapted and optimized in future work”. However, how could anyone decide to change the value of $\sigma_a^2$ when this could possibly lead to vanishing/exploding gradients (at least close to initialization, if I understood correctly)?

**>>Clarification:**
- Line 049-051: it seems to state that it is known that there is a frontier between vanishing-gradient and exploding-gradient that one needs to fit the training in. Do you have a reference for it? How is it known (e.g., from previous literature) that “spurious high-frequency artifacts” can lead to both vanishing and exploding gradients during training?
- Line 051: it is said that we seek a regime where “gradients remain stable”. Does “stable” mean that the norm of the gradient will be uniformly bounded below and above by a non-zero constant?
- Figure 1: for each vertical bar, what does the difference between the two colors (dark vs light gray) represent?
- In the example described in the paragraph starting at line 345: how many input points are used for training?

**>>Other clarifications:**
- About Section 1.2, first paragraph: it is said that SIREN has the tunable parameter $w_0$ to control the frequency range of the network—however, this is only guaranteed at initialization, nothing is known during training, correct? (Unless you are in the NTK domain). Please, clarify this in the text.
- About Section 1.2, first paragraph: it is said that SIREN is related to other alternatives such as positional encoding and Random Fourier features. Why would somebody use SIREN instead of these other alternatives? Is it all about the calibration of $w_0$? If so, why is this so important compared to other approaches that allow for frequency control?
- In Theorem 3.1, clarify in the last sentence that the convergence is as $L \to \infty$.

**>>Other things:**
- Line 039: please, provide examples to better understand “signals of high-frequency content”. I imagine learning functions that correspond to images with considerable texture could be one example, is that correct?
- Please, specify that you consider the $l_2$-norm, e.g., in equation (4).
- In line 254: indicate after $W_l$ that $2\leq l \leq L$.
- For consistency, use “decay” instead of “converge” when referring to the small eigenvalues below line 377.
- Please, proofread the paper for typos and grammatical errors which **should have not happened** at this point. Examples:
  - Delete “have” and the “-d” at the end of “converged” in the last line of page 4 (below line 215).
  - Add parenthesis after “Hinton (2010)” in line 242. Also, “Nair & Hinton (2010)” should be “(Nair & Hinton, 2010)” since it is referring to the paper itself, not the authors as subjects. For this, we can use “\citep” in Latex. More instances of this mistake are found throughout the paper.
  - Title of section 3.1: it says “PRE-CTIVATION”, it should be “PRE-ACTIVATION”.
  - end of line 254: it should say “$W_1$ is sampled” instead of “let $W_1$ be sampled”.
  - line 255: add “and” before “$b_l$”.
  - The first word in Remark 3.2 should be “As”.
  - Line 290: the beginning should be “Under the same assumptions as Theorem 3.1,"

**Questions:**

Please, see the Weaknesses section for most questions. One additional question:
- What happens if in equation (7) one decides to keep the bias uniformly sampled (as in equation (6)), instead of Gaussian sampled, while keeping the rest of equation (7) the same. I wonder what kind of performance will result from it—it could be a good ablation to try.

---

> ### Author Response · Authors · 2025-11-24
> **First Reply - Part 1**
>
> ### **Weaknesses**
>
>
> - *It is my  [...]  larger networks deteriorate?*
>
>   - **Response:** To properly tackle your remark we added an ablation study (p. 28–29) were we compare the performances of our architecture for several depth, and we did not observe such tipping point behavior, with constant network performances. This could be expected since all the results we have are in infinite depth limit.
>     Also we think that it is important to highlight that our initialization is not based on the NTK, we only studied the consequences on a idealized NTK training, to show that our initialization has interesting properties, however all the results derived are just based on the assumption  $N, L \to \infty $, with $N,L$ the width and the depth of the network.
>
> - *The meaning of “limit of large $N$” [...] what makes $N$ large?*
>
>   - **Response:** Thank you for this remark. We changed our notations, and have clarified in the theorem statement that our approach is derived in the limits $N, L \to \infty$. We also reformulated the theorem. The $1/N$ scaling is natural, since these quantities correspond to entries of Jacobian matrices: when such a matrix multiplies another normalized vector, the factor $1/N$ is required to ensure that the resulting vector remains normalized, if you want more detailed about this we included a derivation of the gradient scaling in the appendix (p. 19–20) that explicit this reasoning.
>
> - *Is equation (12) also under the “large $N$” limit?*
>   - **Response:** Fixed
>
> - *Theorem 3.2 holds under random initialization [...] under what probability?*
>   - **Response:** We are not entirely sure what is meant by “the probability under which these results hold.” We hope that our previous clarification addresses this point. For practical networks with finite width $N$ and depth $L$, there is indeed a small probability that a particular random initialization deviates significantly from typical behavior, potentially leading to poor performance. However, by large deviation theory, such extreme events occur with exponentially small probability, with a rate that scales with the network width.
>
> - *The contributions claim results “valuable  [...] what other cases?*
>
>   - **Response:** These results apply to any network using sinusoidal activations, although such activations have so far been used almost exclusively in INR settings. We focused on INR tasks because they already form a broad and important research area that encompasses several widely studied problems.
>
> - *L175–178: why is differentiability important?*
>
>   - **Response:** When dealing with physical signals, it is important to ensure several orders of differentiability, since the derivatives of a signal often have physical meaning. For example, if a neural network encodes a trajectory $x(t)$, one may also need access to the velocity $x'(t)$ or the acceleration $x''(t)$.
>
> - *Paper lacks INR/SIREN applications [...] motivation could be stronger.*
>
>   - **Response:** Since it is a remark shared by all reviewers, we added more application examples (audio fitting, synthetic signals, video fitting, and denoising, p. 23–27). All those new applications are presented in Appendix B of the revised version. We also highlighted that our method outperforms all others on generalization tasks when using deep neural networks. Moreover, we plan to include PDE-related applications in the coming days.
>
> - *Section 4 assumes constant NTK during training [...] please justify.*
>
>   - **Response:** Your observation is correct: this assumption holds only in the infinite-width limit. A proper justification can be found in the foundational work of Jacot et al., *Neural Tangent Kernel: Convergence and Generalization in Neural Networks*, which we now cite in the text. As you noted, the NTK is not constant during training, but this does not affect our initialization, which does not rely on that assumption. Only our analysis of the NTK dynamics under this initialization is restricted to the beginning of training, during the first iterations.
>
> - *Last paragraph of Section 5: how can $\sigma_a^2$ be changed without risking exploding/vanishing gradients?*
>
>   - **Response:** In the revised version, we explained this point in greater detail to make it clearer for readers all along the paper. The exploding/vanishing gradient behavior is controlled solely by \(\sigma_g\). Therefore, fixing $\sigma_g^2 = 1$ imposes only one constraint on the two parameters $c_w$ and $c_b$. This leaves room to impose an additional constraint on $\sigma_a$ in order to enforce other desirable properties. In the refined version of the paper, for instance, we show that choosing $\sigma_a = 0$ yields a bounded Fourier spectrum, which motivates our proposed initialization (p. 7–8).

---

> ### Author Response · Authors · 2025-11-24
> **First Replay - Part 2**
>
> ### **Weaknesses**
>
> - *L049–051: reference for frontier between vanishing/exploding gradients? What about high-frequency artifacts?*
>
>   - **Response:** The frontier you are referring to is known as the edge of chaos (EoC). There is extensive literature on this topic; for example, see *Mean Field Residual Networks: On the Edge of Chaos* by Greg Yang and Samuel S. Schoenholz.
>     Regarding the high-frequency artifacts, these are primarily caused by gradient explosion, which is present in many INR methods. As highlighted in our article, this phenomenon leads to an unbounded Fourier spectrum as network depth increases, caused by not normalized layerwise Jacobian or too important non-linearities. We have added a dedicated section discussing this point in detail (p. 7–8).
>
> - *L051: what does “stable gradients” mean?*
>
>   - **Response:** Stability of the gradients means that the variance of the Jacobian entries remains of order \(1/N\), ensuring that variance is preserved during backpropagation.
>
> - *Figure 1: meaning of dark vs. light gray?*
>
>   - **Response:** The light gray color represent the stantard deviation of the error and the dark color the error, we corrected that in the new version (l. 71).
>
> - *Example starting at l.345: how many input points used?*
>
>   - **Response:** Figure 3, which shows the experimental standard deviation of the pre-activation distribution (left) and of the layer-wise Jacobian entry distribution (right), was not obtained using any training procedure. All the concepts studied here concern initialization, ensuring that training does not begin with ill-conditioned gradients or frequencies. To estimate these distributions, we used 500 input points, as now described in the main text (l. 354).
>
> - *Section 1.2: $w_0$ only controls frequencies at initialization [...] clarify.*
>
>   - **Response:** This is exactly that, and we found furthermore that this control on frequency is ill-conditionned for most architectures when the network is not properly initialized. For example if one starts the training with too high frequency that cannot be encoded in the dataset, aliasing problem will appear. We added a clarification line (l. 377).
>
> - *Section 1.2: why use SIREN instead of PE/FF?*
>
>   - **Response:** Fourier Features (FF) and positional encoding (PE) are fully viable solutions and indeed allow one to bypass spectral bias. However, in our experiments they yield significantly worse performances compared to other architectures, and the literature currently lacks a detailed analysis of how additional layers applied after FF/PE affect the resulting spectrum.
>     For SIREN networks, we showed that in the case of signal encoding, the parameter $w_0$ should be chosen as the Nyquist frequency of the signal—independently of network depth—when using our proposed initialization, which yields an easy calibration to use in real cases, which is not the cases for other methods.
>
> - *Theorem 3.1: clarify convergence as $L\to\infty$.*
>
>   - **Response:** Fixed
>
> - *L039: example of “high-frequency content”?*
>
>   - **Response:** Yes it is a really good example, we added added a clarification l.39
>
> - *Specify $l_2$-norm in Eq. (4).*
>
>   - **Response:** Fixed
>
> - *L254: indicate $2\le l\le L$ after $W_l$.*
>   - **Response:** Could you clarify this, we precised the range of l before the definition of $W_l$
>
> ## Questions
> - What happens if in equation [...] good ablation to try.
>   - **Response:** What you suggest is very interesting, but it would break our theoretical results, since the pre-activations would no longer be Gaussian but a superposition of a uniform and a Gaussian distribution. We therefore leave such experiments for future work, when we study dynamical isometry and the corresponding constraints on the weight and bias distributions.

---

### Official Review · Reviewer_s653 · 2025-10-30

**Soundness:** 2
**Presentation:** 2
**Contribution:** 2
**Rating:** 4
**Confidence:** 4

**Summary:**

\paragraph{Summary}
The paper proposes a closed-form initialization for SIREN-style sinusoidal networks. Hidden-layer weights and biases are set via two scalars \((c_w, c_b)\) chosen so that pre-activations converge to a fixed variance and the layerwise Jacobian entry variance satisfies
$
\sigma_g^2 = \frac{1}{N}.
$
Solving these constraints yields
$
c_w = \sqrt{\frac{6}{1+e^{-2}}}, \qquad
c_b = \sqrt{\frac{1}{3}}\, c_w e^{-1}
\quad \text{(Eqs.~(7)--(8)).}
$
Experiments on 1D/2D/3D synthetic functions and a single image-fitting task suggest improved “generalization” (via upsampled reconstructions) over baselines. NTK measurements indicate the mean NTK eigenvalue grows roughly linearly with depth under the proposed initialization, versus exponential growth.

**Strengths:**

- A simple, analytical initialization for sine activations; easy to implement (Eqs. (7)–(8)).

- Clear fixed-point analysis for pre-activation variance using Lambert-W; explicit gradient-variance target.

- Useful NTK-based depth scaling narrative; the paper measures trace/eigenvectors and links depth to speed.

- Repeated small-scale experiments show consistent wins on their metrics.

**Weaknesses:**

- The Jacobian-variance factorization assumes $W_\ell \perp z_\ell$ even though $z_\ell = W_\ell h_{\ell-1} + b_\ell$. Without a large-width justification (e.g., leave-one-out / tensor-program) the claim that targeting $\sigma_g^2 = 1/N$ stabilizes gradients is fragile.
- Matching entrywise Jacobian variance does not ensure favorable singular-value spectra of end-to-end Jacobians (dynamical isometry). No SVD/condition-number results are reported.
- No correlation recursion or $\chi$-coefficient/fixed-point correlation study to substantiate an EoC claim beyond gradient-variance targeting.
- Mostly small synthetic INR tasks and a single image; no NeRF/PINN or noisy/irregular sampling settings. Image evaluation uses upsampled reconstructions instead of standard PSNR/SSIM.

**Questions:**

- Can you justify or bound $\mathrm{Cov}(W_{\ell,ik}, z_{\ell,i})$ as width $N$ grows? A tensor-program or leave-one-out derivation—or empirical covariance vs.\ $N$—would strengthen Theorem~3.2.
 - Please report singular-value distributions (or spectral norms/condition numbers) of end-to-end Jacobians across depths and widths for multiple inputs, to demonstrate stability beyond entrywise variance.
- Beyond the trace, provide the NTK eigenvalue spectrum vs.\ depth (including tails and condition numbers) and analyze eigenvector frequency alignment to support claims about depth-wise learning dynamics.
 - Add kernel-bandwidth SIREN inits and residual/skip/normalized SIREN variants. Include realistic INR settings (e.g., NeRF or PINNs) with standard metrics (PSNR/SSIM/relative $L_2$).
- Evaluate under noisy/irregular sampling, distribution shift, and different input scalings to test whether the fixed-point targeting remains effective.
- Will you release code, seeds, and exact training logs? A concise “How to initialize” box in the main text would aid adoption.

---

> ### Author Response · Authors · 2025-11-23
> **First Reply**
>
> # Reviewer 2
>
> ### **Weaknesses**
>
> - *The Jacobian-variance factorization assumes $W_\ell \perp z_\ell$ [...] stabilizes gradients.*
>
>   - **Response:** Thanks for this precision, in Theorems~3.1-3.2 we take the limit of large width which automatically yields the independance of $W_{\ell,ik}$ and  $z_{\ell,i}$ by a leave one out derivation, we dont need to suppose the independance, this hypothesis is traditionally done in initialization schemes see (Xavier-Glorot), this is why we added it.
>
> - *Matching entrywise Jacobian variance does not ensure favorable singular-value
> spectra [...] no results reported.*
>
>   - **Response:** This is true. We focused only on the variance of the Jacobian entries and the pre-activation variance, not on deeper properties of the end-to-end Jacobian, such as a unitary singular value spectrum. We leave this aspect for future work. This has been added it to the conclusion / Perspective section, mentioning a new figure in the Appendix (see p.20,21), showing and discussing singular values of the end-to-end Jacobian matrix.
>
> - *No correlation recursion or $\chi$-coefficient/fixed-point study [...] EoC claim.*
>
>   - **Response:** A classical EoC claim follows from studying the fixed point of $q$, the pre-activation variance, together with the fixed point of $ \chi = \frac{1}{N} \langle Tr (J_l^{T} J_l) \rangle,$
>     which represents the mean of the singular value spectrum of the $\ell$-th layer Jacobian. When $\langle J_\ell \rangle = 0$, one can show that $\chi = N \tilde{\sigma}_l^2$ , with the layer wise jacobian variance at layer $\ell, \tilde{\sigma_l}^2$. In our paper, we identify fixed points for both $\sigma_l^2 = q$ and $\tilde{\sigma_l}^2$, which is therefore equivalent to establishing an EoC claim.
>
> ### **Questions**
>
> - *Can you justify or bound  [...] strengthen Theorem 3.2?*
>
>   - **Response:** We precised the leave-one out derivation to show that $z_\ell$ and $ W_\ell$ are asymptotically independent.
>
> - *Please report singular-value distributions [...] demonstrate stability beyond entrywise variance.*
>
>   - **Response:** We reported in the appendix (p.20-21) the full singular value distributions of the end-to-end Jacobian $J$. Our results show that the proposed initialization yields a normalized, depth-independent $s_{\max}$, but it does not produce a fully unitary singular value spectrum. This leaves open the question of achieving dynamical isometry in this class of neural networks, which we consider an interesting direction for future work.
>
> - *Beyond the trace, provide NTK eigenvalue spectra vs. depth [...] support depth-wise learning claims.*
>
>   - **Response:** In the appendix, we provide the detailed eigenvalue spectra (page 21) for multiple depths and several initialization schemes. We observe that the spectrum of our proposed initialization remains largely stable as depth increases, in contrast to other schemes, which supports our NTK analysis.
>
>     Additionally, we include the power spectrum of the eigenvectors (p.22) to study the correspondence between eigenvectors and frequencies. For our initialisation, the mapping is nearly identity for frequencies below $w_0$, which further corroborates our Fourier analysis presented in the added section.
>
> - *Add kernel-bandwidth SIREN inits and residual/skip/normalized variants [...] include realistic INR settings.*
>
>   - **Response:** As requested by Reviewer~1, we added several state-of-the-art methods—FINER, WIRE, and ReLU(PE). For readability, we do not include additional initialization schemes. It is worth noting, however, that all SIREN initializations proposed so far lead to exploding gradients in the limit of large depths, resulting in very poor performance for deep networks.
>
>     Regarding realistic INR settings, we applied our method to audio fitting, image fitting, video fitting, and denoising. As mentioned to Reviewer~1, we also plan to include a PINN comparison in the coming days.
>
>     Finally, we have added the requested standard metrics (SNR / SSIM) highlighting the good performance of the proposed scheme compared to other initialization schemes.
>
> - *Evaluate under noisy/irregular sampling, distribution shift, and input scaling [...] test fixed-point robustness.*
>
>   - **Response:** For the simple synthetic tasks (page 26-27), the sampling was randomized, and we additionally provide denoising experiments (page 25) in the appendix. Concerning different input scalings, we refer to the audio fitting task performed on a 7-second, 44100 Hz signal (page 23), as well as the planetary video fitting (page 24) conducted on the Earth’s (longitude × latitude) domain. If you are interested in exploring the limits of our theoretical framework, we recommend the ablation study included in the appendix (page 28-29).
>
> - *Will you release code, seeds, and logs? [...] aid adoption.*
>
>   - **Response:** All the code will be released with exact training logs, and we will add a "How to initialize" box in appendix.

---

### Official Review · Reviewer_K6uc · 2025-10-31

**Soundness:** 3
**Presentation:** 3
**Contribution:** 3
**Rating:** 6
**Confidence:** 4

**Summary:**

This work introduces a theoretically grounded initialization scheme for SIRENs that enforces:
 (1) the preservation of the pre-activation distribution across layers, and
 (2) stable gradients throughout the network (avoiding vanishing or exploding gradients).

The authors evaluate their method on image-fitting tasks and demonstrate stable training in deep SIRENs. Furthermore, they leverage Neural Tangent Kernel (NTK) theory to show that the proposed initialization avoids the introduction of spurious high-frequency components at the start of training.

Training deep SIRENs is a challenging problem with several applications in implicit neural representations (INRs). I tested the proposed initialization on image-fitting experiments and found it promising. However, a straightforward extension to SDF fitting did not yield good results, suggesting that this adaptation may not be trivial.

**Strengths:**

The paper is clearly written and well organized.

The proposed initialization is supported by NTK theory, which provides a solid theoretical foundation.

The empirical results indicate robustness in fitting tasks.

Improving the trainability of deep SIRENs has significant implications for a wide range of INR applications.

**Weaknesses:**

[L53–54] It is unclear whether the authors refer to (1) the gradient of the loss function with respect to network parameters or (2) the derivative of the network output with respect to input coordinates. In L212, it seems to refer to (2). This point is addressed in Eqs. 10–11.
Could the authors include a specific example where gradients explode under the standard SIREN initialization but remain stable with the proposed one? In Fig. 1, both methods seem to yield noisy reconstructions. Including analytical gradient visualizations (as in TUNER) would provide a clearer comparison.

[L107] TUNER [1] is a closely related work proposing frequency-aware initialization for SIRENs. It also offers a formula that could explain the emergence of high-frequency artifacts when increasing network depth [L43]. I recommend including it in the comparisons. Additionally, FINER [3] is another recent method addressing similar initialization and spectral control challenges.

[L132] Evaluation is restricted to RGB image fitting with an MSE loss (Eq. 4).

[L144] INRs are generally trained to represent low-dimensional signals (e.g., 2D images, SDFs, occupancy, or radiance fields), often including a regularization term (e.g., the Eikonal constraint for SDFs).
It would strengthen the paper to show results for SDF fitting or at least discuss how the proposed initialization behaves under such regularization. I attempted to implement the initialization for SDF reconstruction using the repository from “Exploring Differential Geometry in Neural Implicits” and observed poor convergence.

[L90] Yüce et al. [2] should be cited, as they also analyze SIRENs through the NTK framework.

[L159] Mention explicitly that \theta denotes the union of all \theta_i​’s.

[Fig. 2] A 10-layer MLP with 256 neurons per layer is quite large for image fitting. Testing smaller architectures (e.g., 16 or 32 neurons) would help assess scalability.

**Questions:**

[L25] When stating that the new initialization “consistently outperforms state-of-the-art methods,” which methods are considered SOTA? If only Xavier, Kaiming, and SIREN, this comparison is limited, these are not typically used for signal representation. Please also consider TUNER [1], FINER [3], and [4].

[L47] Does the proposed initialization accelerate training convergence?

How does the method perform on high-frequency audio fitting, where deeper architectures might be more beneficial?

[L188] Does the initialization assume identical hidden-layer widths (same N) across the network?


[1] Novello et al., Tuning the Frequencies: Robust Training for Sinusoidal Neural Networks, CVPR 2025.

[2] Yüce et al., A Structured Dictionary Perspective on Implicit Neural Representations, CVPR 2022.

[3] Liu et al., FINER: Flexible Spectral-Bias Tuning in Implicit Neural Representation by Variable-Periodic Activations, CVPR 2024.

[4] Yeom et al., Fast Training of Sinusoidal Neural Fields via Scaling Initialization, ICLR 2025.

---

> ### Author Response · Authors · 2025-11-23
> **First Reply**
>
> A global detail of all the change of this refined version has been written as comment of the main text.
>
> ### **Weaknesses**
>
> - *[L53–54] It is unclear [...] a clearer comparison.*
>
>   - **Response**: We added a clearer comparison using gradient visualisation and emphasised from the beginning of the article that our initialisation provides gradient control over both the parameters and the input. You also pointed out that our proposed initialization remained noisy despite this gradient control. This stems from our initial assumption that a normalized gradient would prevent the emergence of spurious high–frequency components as depth increases, an assumption questioned by Reviewer 4.
>     We added a new section on the Fourier spectrum of sinusoidal neural networks (see p.7). This leads to an alternative initialisation that contradicts the Sitzmann $\sigma_a = 1$. Instead, we propose $\sigma_a = 0$, yielding a depth–independent Fourier spectrum and removes the spurious high–frequency noise observed beyond the Nyquist limit.
>
> - *[L107] TUNER [1] is an [...] control challenges.*
>
>   - **Response:** It indeed proposes an interesting way to limit and derive the Fourier spectrum of such networks, and we intend to further explore this training regularisation that enables such control. Regarding comparisons, we did include FINER and WIRE as other state-of-the-art baseline (see p.5). However, we did not tackle TUNER [1] because it requires manually selecting the first-layer frequency, which does not align with our perspective—namely, fixing $w_0$ to the Nyquist frequency of the sampling. Nevertheless, we strongly consider including a comparison with TUNER in future work.
>
> - *[L132] Evaluation [...] with an MSE loss (Eq. 4).*
>
>   - **Response:** Our benchmark (Fig. 1) included synthetic INR tasks (such as 1d, 2d and 3d  multi-frequencies function fitting, p.26-27), with additional details provided in the appendix.  Further comparisons for audio (p.23) and video (p.24) fitting have been included as well. Finally, we plan to demonstrate improved performance in PINN reconstructions in upcoming experiments in the following days.
>
> - *[L144] INRs are generally [...]poor convergence.*
>
>   - **Response:** SDFs offer an interesting perspective to explore; however, the final goal of our research  will be the study of PINNs, which rely on a similar form of regularized reconstruction. In this context, the choice of $w_0$ becomes much less clear. Nevertheless, normalising the gradient of such networks consistently yields useful properties, as it prevents the emergence of large gradient values, which could otherwise make the optimisation stiffer. From this perspective, we plan to include a PINN comparison in the coming days.
>
> - *[L90] Yüce et al. [2] [...] NTK framework.*
>   - **Response:** Again this article is really interesting, since it lays the emphasize on aliasing problem in the case of overfitting that w specifically want to avoid. We did not mention it extensively in the ntk section since it deals with meta-learning justification that we do not study here. Furthermore, we added a reference for their work on aliasing (p.7).
>
> - *[L159] Mention explicitly [...] *
>
>   - **Response:** Fixed
>
> - *[Fig. 2] A 10-layer MLP [...] assess scalability.*
>
>   - **Response:** We included in the appendix an ablation study where we tested smaller architectures, and observed the same spectral behavior, p.28,29)
>
> ### **Questions**
> - *[L25] When stating that the new  [...] and [4].*
>
>   - **Response:** We consider your remarks and added the FINER [3], WIRE and ReLU(PE), we did not include [4] since it needs a previous finetuning on the hidden-weight rescaling (we consider implementing this method with our initialisation in future work)
>
> - *[L47] Does the [...] convergence?*
>
>   - **Response:** As explained in [2] overfitting methods that are common in the literature, yields rapid convergence due to high frequency embedding. In this article we showed that this is also about having large gradient variance, that is rescaling the NTK eigenvalues leading to a rescaling of the learning rate. So in term of training convergence, it does not have the same performance as a traditional SIREN which is able to use higher frequency to reduce the training error, but regarding training speed to reach the test error, it is experimentally the same
>
> - *How does [...] might be more beneficial?*
>   - **Response:** We tested our network on audio fitting (p.23), and regarding the generalization error it outperforms well the selected SOTA method for large depth networks. The associated experiments have been added in Appendix B.3
> - *[L188] Does the initialisation assume [...] across the network?*
>   - **Response:** The theory requires layers of infinite width. For simplicity of notations, we assumed a constant width $N$ for every layer, but allowing each layer to have its own number of neurons $n_\ell$ does not affect the proof in the limit $n_\ell \to \infty$.

---

### Author Response · Authors · 2025-11-23
**First refinements after reviewers comments**

**We thank the reviewers for their constructive and insightful comments** . Below, we provide a detailed response to all remarks and suggestions. In the revised version of the manuscript, we have :
- Added a discussion on frequency spectrum of the network, addressing the main questions raised by Reviewers~1 and~4. Focusing on constraints from frequency (page 7) propagation (rather than solely on the activation function at the fixed point) also led us to propose an even simpler and more powerful initialization with weight variance equal to $\sqrt{3}$, which is now discussed in the text.
- Included additional metrics confirming that the proposed initialization operates at the edge of chaos, following common practice in the recent literature (as requested by Reviewer 2). Notably, this line of work initially focused on classification problems rather than INR, suggesting a potentially interesting role for sine activations in that context as well.
- Added more application examples, as requested by Reviewers 1, 2, and 4, including fitting of 1D and 2D signals, videos, geophysical images, and audio (page 23-29). We also provide comparisons with other recent methods (e.g., FINER, WIRE), as suggested by Reviewer~1.
- Clarified the assumptions underlying our derivations and the NTK framework (Reviewer 3), and added more details on the NTK matrix (Reviewers 2 and 3, page 22-23).

Addressing these points led to substantial modifications of the original manuscript. To keep the 10-page format, most of the new results are presented in the appendices. We provide a .pdf version of the revised manuscript with changes highlighted in color.

---

### Public Comment · ~Simon_Kuang1 · 2025-11-26
**Uniform(-π, π) biases for stable activations?**

There is another way to initialize a sinusoidal neural net which exactly fixes the pre-activation and gradient variances.

- Initialize all weights to Uniform(-π, π). Now the preactivation $z = Wh + b$ is uniformly distributed on the torus, irrespective of the distribution of $h$.
- This ensures that each layer's initial distribution is $\sin(z)$, consisting of independent entries with with mean 0 and variance 1/2.

By induction, this also stabilizes the preactivation variances and it stabilizes the layer gradient variances too: by initialization, $h$ has independent entries with with mean 0 and variance 1/2. Then, if the entries of $W$ are $\mathcal N(0, \sigma^2_w)$, we have

$$
\begin{align}
\nabla_h \sin(Wh + b)
&= \operatorname{diag}\cos(Wh + b) W
\\\\
\operatorname{cov} \nabla_h \sin(Wh + b)
&= \mathbb E \operatorname{diag}\cos(Wh + b) W W^\intercal \operatorname{diag}\cos(Wh + b)
\\\\
&= \frac{\sigma^2_w \ \text{fan-in}}{2} I
\end{align}
$$

This differs from your and SIREN's initialization by virtue of the large bias term.
In spite of seemingly ideal pre-activation and gradient scaling, I have found that deeper networks initialized in this way tend to be very hard to train.
So my experience would suggest that having nice variances is necessary, but unlike in e.g. ReLU, not sufficient for initial training. Does your frequency-domain analysis shed light on this puzzle?

---

> ### Author Response · Authors · 2025-11-27
> **First Reply**
>
> - In our paper, we show that for any Gaussian bias and uniform weight initialization, the variance of the *pre-activations* and the variance of the *Jacobian entries* converge to a **fixed point** (Theorems 3.1 and 3.2).
>
> - In your setting, if you replace the uniform bias distribution with a Gaussian one, you will find that the Jacobian entries are no longer normalized (see Figure 3). In this regime, the backpropagation signal grows exponentially with network depth, making the gradient (or end-to-end Jacobian) extremely ill-conditioned:
>   $$
>   J = J_{1} J_{2} \dots J_{L}.
>   $$
>   As a result, training deep networks becomes complicated, and the learned function exhibits excessively large derivatives.
>
> - We also believe your calculations rely on the Sitzmann approximation $\text{Var} [\sin(z)] \sim 1/2$, but this is not accurate enough (see the proof of Theorem 3.1). Accounting for the exact behavior of the sine function is what leads to the **fixed-point** structure we identify. Moreover, even under this simplified approximation, your initialization does not yield a variance of the Jacobian entries of layer $\ell$, $J_\ell$, exactly equal to $1/\sqrt{N}$ (see the proof of thm 3.2). Thus, even under the simplified picture, the desired normalization is not achieved.
>
> - This is sufficient to show that using your initialisation the training will be impaired. Regarding the frequency study, it is used to add a constraint on the variance of the pre-activation. Indeed, controlling the end-to-end Jacobian is necessary both for stable training and for governing how “wiggly” the final network becomes. However, we also showed that this control is **not equivalent** to controlling the Fourier spectrum of the network: even when the end-to-end Jacobian is normalized, nonzero pre-activations introduce increasingly high-frequency components with depth. This observation motivates our choice of $\sigma_a = 0$, which ensures a depth-invariant Fourier spectrum.

---

> ### Public Comment · ~Simon_Kuang1 · 2025-11-27
>
> Thank you for your reply.
>
> The motivation for my question is: why would I like the pre-activation and Jacobian variances to _converge_ to a fixed point when I can simply set them by fiat?
>
> > In your setting, if you replace the uniform bias distribution with a Gaussian one,
>
> Fair, but my questioning is all about Uniform(-pi, pi) biases.
>
> > We also believe your calculations rely on the Sitzmann approximation
>
> No, Sitzmann has several layers of approximation. I am using the following exact calculation. If $z$ is uniformly distributed on the unit circle, then $\text {Var} \sin(z) = \text{Var} \cos (z) = 1/2$ exactly. One way to derive this is by taking the $\sigma^2 \to \infty$ limit of your Lemma A.2. Another way is by integrating using the uniform measure on $z \in [0, 2\pi]$.
>
> Moreover, it is possible to force $z$ to be Uniform on the unit circle by initialization. Note that if $\theta_0$ is _any_ random variable (here, representing the scaled activation from the previous layer) and $\theta \sim \text{Unif} (-\pi, \pi)$ is independent (representing the random bias), then $\theta_0 + \theta$ is Uniform modulo $2\pi$. Do you agree with this claim?
>
> I agree with your discussion on the Jacobian scaling. The spirit of my question is: why are there training difficulties with $\text{Unif} (-\pi, \pi)$ initialization even when the Jacobian is stable?

---

> > ### Author Response · Authors · 2025-11-28
> > **Second Reply**
> >
> > We agree with the modulo $2\pi$ statement: if $\theta \sim \mathrm{Unif}(-\pi,\pi)$ and $\theta_0$ is any independent random variable, then $(\theta_0+\theta)\bmod 2\pi$ is exactly uniform on the circle. Consequently, for the wrapped variable $z\bmod 2\pi$ one has
> > $$
> > \mathbb{E}[\sin z]=0,\qquad \text{Var}(\sin z)=\frac12,
> > $$
> > independently of the distribution of $\theta_0$. Our analysis, however, tracks the distribution of the **real-valued** preactivations $z = Wh + b \in \mathbb{R}$, not only their phase modulo $2\pi$. Under the usual mean-field assumptions $N \to \infty$,  the coordinates of $Wh$ are approximately Gaussian. On the real line, adding a uniform bias $b\sim\mathrm{Unif}(-\pi,\pi)$ gives $z = Wh + b$, whose distribution is the **convolution** of a Gaussian with a uniform variable, which will not be uniform on $\mathbb{R}$.
> >
> > Regarding the Jacobian: even if each per-layer Jacobian has entries whose variance is tuned at initialization, trainability depends on the scaling of the end-to-end Jacobian. In our notation the relevant per-layer gain is $\mathrm{Var}(J_\ell)\sqrt{N}$. For stable gradient flow one needs
> > $$
> > \mathrm{Var}(J_\ell)\sqrt{N} = 1.
> > $$
> > With $\mathrm{Unif}(-\pi,\pi)$ using your calculation on the unit circle (this will be the same on the real-line), this condition is not satisfied so the end-to-end Jacobian scales as
> > $
> > \big(\mathrm{Var}(J_\ell)\sqrt{N}\big)^L,
> > $
> > which explodes with depth in your case, leading to ill-conditioning even when single layers look well behaved.

---

> ### Public Comment · ~Simon_Kuang1 · 2025-11-28
>
> > On the real line, adding a uniform bias $b\sim\mathrm{Unif}(-\pi,\pi)$ gives $z = Wh + b$, whose distribution is the convolution of a Gaussian with a uniform variable, which will not be uniform on $\mathbb{R}$.
>
> I agree.
> However, $z$ is uniform modulo $2\pi$. Thus the distribution of $\sin(z)$ is the same as if $z$ were uniform on $2\pi$, correct? This is the essence of my question. Adding a $\mathrm {Unif}(-\pi, \pi)$ bias effectively fixes the output distribution of $\sin(z)$ irrespective of the distribution of $Wh$.
>
> (In other words, because $\sin$ is periodic, wrapping the distribution of a pre-activation by $2\pi$ does not change the activation distribution.)

---

> ### Public Comment · ~Simon_Kuang1 · 2025-11-28
>
> Another way to see this is by [trigonometric identities](https://en.wikipedia.org/wiki/List_of_trigonometric_identities#Angle_sum_and_difference_identities). Suppose that $Wh$ has **any distribution** (e.g. constant, uniform, or Gaussian) on $\mathbb R$, and $b \sim \mathrm{Unif} (-\pi, \pi)$ independently. Then, the mean of $\sin(z)$ is 0 by:
>
> $$
> \begin{align}
> \mathbb E \sin(Wh + b)
> &= \mathbb E \left(\sin(Wh) \cos(b) + \cos(Wh) \sin(b) \right)
> \\\\
> &= (\mathbb E \sin(Wh)) (\mathbb E \cos b) + (\mathbb E \cos(Wh)) (\mathbb E \sin(b))
> \\\\
> &= (\mathbb E \sin(Wh)) (0) + (\mathbb E \cos(Wh)) (0)
> \\\\
> &= 0
> \end{align}
> $$
>
> and the variance is 1/2 by:
>
> $$
> \begin{align}
> \mathbb E \sin^2(Wh + b)
> &= \frac{1}{2} \mathbb E (1 - \cos(Wh + b))
> \\\\
> &= \frac{1}{2} \mathbb E (1 - \cos(Wh) \cos(b) - \sin(Wh) \sin(b))
> \\\\
> &= \frac{1}{2} (1 - \mathbb E \cos(Wh) \mathbb E \cos(b) - \mathbb E \sin(Wh)\mathbb E \sin(b))
> \\\\
> &= \frac{1}{2}
> \end{align}
> $$

---

> > ### Author Response · Authors · 2025-11-30
> > **Third Answer**
> >
> > Indeed, what you said is exactly right. If you choose the bias from a uniform distribution $b \sim \mathcal{U}[-\pi,\pi]$, then the distribution of the resulting layer outputs is fixed to the “$\text{arcsin}$” distribution. By then setting $\sigma_w$ appropriately, you can obtain exact control over the variance of the Jacobian entries (using your previous calculation, we see that for a uniform distribution this recovers the Sitzmann weighting).
> >
> > So you were right from the start: our discussion of the Fourier spectrum precisely explains why, even if we take the correct weight scaling, the increase in high-frequency due to the non-vanishing non-linearities will impact our generalisation abilities.
> >
> > To put it in a nutshell: with your initialization, you obtain a constant output distribution for each layer, which is a very nice property, thanks for pointing that out. And with the Sitzmann scaling for uniformly distributed weights, you get exact control over the Jacobian entries. However, the underlying frequency problem remains unsolved. This is very close to the case $\sigma_a = 1$ in our paper, for which convergence to the fixed point is exponentially fast, so we obtain the same property.

---

### Public Comment · ~Simon_Kuang1 · 2026-07-08
**Reproduction code**

Can the authors share reproduction code for the camera-ready version?

edit: from scanning the poster's QR code, I have arrived at [this GitHub repository](https://github.com/AndreaCombette/SIREN-Init).


While the Reproducibility statement reads,

> All source code used in our experiments is provided in the supplementary material, including implementations of the architectures used for comparison,

this repository contains essentially the same content as the supplementary material and omits most of the experiments presented in the paper.

**In particular, Figures 3–7, 9–16, 20–21 are not reproduced.**

---

> ### Public Comment · ~Simon_Kuang1 · 2026-07-16
> **$$c_w$$**
>
> Moreover, I am concerned that there the reproduction repository differs in a small but material way from the paper. The paper proposes exact formulas for $c_w$ and $c_b$ (equation 9). However, [the provided code](https://github.com/AndreaCombette/SIREN-Init/blob/main/src/architecture.py#L65) does not use the numerical value of $c_w$ in the paper ([permalink](https://web.archive.org/web/20260716214743/https://github.com/AndreaCombette/SIREN-Init/blob/main/src/architecture.py)).
>
> The "EOC" code path reads:
>
> ```py
> bound = self.c / math.sqrt(self.in_features) / self.omega_0
> self.linear.weight.uniform_(-bound, bound)
> var_w = np.var(self.linear.weight.data.numpy()) * self.in_features
> bound_bias = np.sqrt(1 - 0.5 * var_w * (1 - np.exp(-2)))
> ```
>
> The code uses the sample variance `var_w` instead of the population variance `c_w`.

---

### Meta-Review · Area_Chair_yMx9 · 2025-12-26

**Summary:**

The reviewers had the following concerns:
1. Insufficient empirical validations. Specifically, the initial manuscript is missing 1) experiments from different task domains, 2) comparison with other existing frequency-aware initialization schemes, 3) ablation studies of the proposed initialization under varying network scale and depth, and 4) experiments that show the spectral properties of the Jacobian matrix under the proposed init. scheme.
2. Claims based on unjustified causal relationships. The initial proposed scheme aims to suppress the spurious high-frequency modes by controlling the gradient norms at initialization, which is not necessarily effective, as pointed out the some reviewers.
3. Clarity issue in the theoretical statements. Reviewers have asked several clarification questions regarding, for example, the independence between weights and inputs at each layer, the interpretation of Theorem 3.2, etc.

**Reviewer Concerns:**

The authors have provided a large set of experiments to address the reviewers' concerns, covering most of the suggestions in their reviews. I have briefly checked most of the new experiment results. Although not all reviewers would find those results fully satisfactory, most of them are still promising. Thus, I consider the first concern mostly addressed.

Likewise, the authors have provided a new init. scheme (under the same theoretical framework) that targets vanishing pre-activation variance at the late layers, to suppress the high-frequency modes. I understood the rationale behind this new proposal, and it seems to have better empirical performance. Thus, I consider the second concern addressed.

I also consider the last concern largely addressed since the authors' responses to those technical questions make sense to me.

**Reviewer Scores:**

If the reviewers had participated fully in the discussion, I would imagine some score changes towards the positive side. At least one of the reviewers (s653 and Le3d) would likely increase their scores (from 4 to 6). I am less certain about Reviewer 1Ux1 (with score 4). I think Reviewer K6uc would keep their score (6).

Overall, the hypothesized "post-rebuttal" score changes would push the paper slightly above the borderline. Since the added experiments look promising to me, I would also lean towards acceptance.

---

### Decision · Program_Chairs · 2026-01-26

Accept (Poster)